



# Development of the DRoplet Ice Nuclei Counter Zürich (DRINCZ): Validation and application to field collected snow samples

Robert O. David[1,*], Maria Cascajo Castresana[1,2], Killian P. Brennan[1], Michael Rösch[1], Nora Els[4], Julia Werz[3], Vera Weichlinger[1], Lin S. Boynton[3], Sophie Bogler[3], Nadine Borduas-Dedekind[1,3], Claudia Marcolli[1], Zamin A. Kanji[1]

[1]Institute of Atmospheric and Climate Science, ETH Zürich, Zürich, 8092, Switzerland
[2]CIC nanoGUNE Consolider, Donostia-San Sebastian, E-20018, Spain
[3]Institute for Biogeochemistry and Pollutant Dynamics, ETH Zürich, Zürich, 8092, Switzerland
[4]Institute of Ecology, University of Innsbruck, Innsbruck, 6020, Austria
*Now at Department of Geosciences, University of Oslo, Oslo, 0315, Norway

*Correspondence to*: Robert O. David (r.o.david@geo.uio.no) and Zamin A. Kanji (zamin.kanji@env.ethz.ch)

**Abstract.** Ice formation in the atmosphere is important for regulating cloud lifetime, Earth's radiative balance and initiating precipitation. Due to the difference in the saturation vapor pressure over ice and water, in mixed-phase clouds (MPCs), ice will grow at the expense of supercooled cloud droplets. As such, MPCs, which contain both supercooled liquid and ice, are particularly susceptible to ice formation. However, measuring and quantifying the concentration of ice nucleating particles (INPs) responsible for ice formation at temperatures associated with MPCs is challenging due to their very low concentrations in the atmosphere (~ 1 in $10^5$ at -30 °C). Atmospheric INP concentrations vary over several orders of magnitude at a single temperature and strongly increase as temperature approaches the homogeneous freezing threshold of water. To further quantify the INP concentration in nature and perform systematic laboratory studies to increase the understanding of the properties responsible for ice nucleation, a new drop freezing instrument, the DRoplet Ice Nuclei Counter Zurich (DRINCZ) is developed. The instrument is based on the design of previous drop freezing assays and uses a USB camera to automatically detect freezing in a 96-well tray cooled in an ethanol chilled bath with an automated analysis procedure. Based on an in-depth characterization of DRINCZ, we develop a new method for quantifying and correcting temperature biases across drop freezing assays. DRINCZ is further validated performing NX-illite experiments, which compare well with the literature. The temperature uncertainty in DRINCZ was determined to be ± 0.9 °C. Furthermore, we demonstrate the applicability of DRINCZ by measuring and analyzing field collected snow samples during an evolving synoptic situation in the Austrian Alps. The field samples fall within previously observed ranges for cumulative INP concentrations and show a dependence on air mass origin and upstream precipitation amount.



## 1 Introduction

In the atmosphere, ice plays an important role in initiating precipitation and affects the radiative properties of clouds. As much as 80% of land falling precipitation initiates through the ice phase (Mülmenstädt et al., 2015), making it essential to understand the pathways for ice formation in the atmosphere. The ratio of cloud droplets to ice crystals in a mixed-phase cloud (MPC) alters the radiative properties of the cloud and its lifetime (Lohmann and Feichter, 2005; Matus and L'Ecuyer, 2017; Tan et al., 2016). This ratio is important for future climate projections as warmer temperatures will lead to a decrease in ice content, ultimately increasing cloud lifetime and cloud albedo (Tan et al., 2016). Additionally, ice formation at temperatures above -38 ˚C in the atmosphere occurs primarily in MPCs through the freezing of cloud droplets (Ansmann et al., 2009; Boer et al., 2011; Westbrook and Illingworth, 2011). Therefore, understanding ice formation in conditions associated with MPCs is of the utmost importance.

When an ice nucleating particle (INP) gets immersed in cloud droplets either by acting as cloud condensation nucleus or through scavenging by cloud droplets, the INP can induce ice formation by reducing the energy barrier associated with the formation of an ice germ and thus freeze at warmer temperatures than homogeneous freezing (Vali et al., 2015). To reproduce the immersion freezing pathway in the laboratory, several methods are used. Single particle methods, such as continuous flow diffusion chambers (Rogers, 1988; Stetzer et al., 2008) operated at water supersaturated conditions (DeMott et al., 2015, 2017; Hiranuma et al., 2015), or with extended chambers that activate individual particles into cloud droplets before exposing them to supercooled conditions (Burkert-Kohn et al., 2017; Kohn et al., 2016; Lüönd et al., 2010) allow for the quantification of the number concentration of INPs as a function temperature. Larger laboratory based single particle methods for examining INPs in the immersion mode include expansion chambers where cloud droplets are first formed by adiabatic cooling due to the expansion of an air volume (Niemand et al., 2012) or experiments where droplets are initially activated and then subsequently cooled as they travel through a laminar flow tube (Hartmann et al., 2011). However, the single particle methods have detection limitations due to the background ice crystal concentration of the chamber and the optical methods for discriminating between ice and water. Due to the rarity of INPs at MPC conditions, single particle methods are typically unable to quantify INP concentrations within natural ambient samples at temperatures higher than approximately -22 ˚C in remote regions or without the use of concentrators (Cziczo et al., 2017).

In contrast bulk methods such as, drop freezing assays (Bigg, 1953; Hill et al., 2014; Stopelli et al., 2014; Vali, 1971), differential scanning calorimetry (Kaufmann et al., 2017; Marcolli et al., 2007) and microfluidic devices (Reicher et al., 2018; Riechers et al., 2013; Stan et al., 2009; Tarn et al., 2018) can be used to detect even the lowest atmospheric INP concentrations. The majority of atmospheric INP concentrations at temperatures above -15 ˚C has been quantified using drop freezing assays. To retrieve the concentrations of INP from such bulk suspensions, Vali, (1971; 2019) showed that by dividing a sample into several aliquots, it is possible to calculate the number of INPs present in the sample as a function of temperature. The



probability for more than one INP in an aliquot that freezes at the same temperature can be predicted using Poisson's Law
(Vali, 1971). Following Vali (1971), the cumulative number of INPs in a given sample for each temperature can be calculated
as:
$$INP(T) = \frac{-ln(1-FF(T))}{V_a} \qquad (1)$$
where $FF(T)$ is the fraction of frozen aliquots at a given temperature, $T$, and $V_a$ is the volume of an aliquot. As can be seen in
Eq. 1, the only way to extend the range of measureable INPs across temperature scales is to change $V_a$. Due to instrumental
limitations, it is often difficult to change $V_a$ by significantly enough values for a change in $INP(T)$ within a single instrumental
setup. Rather it is easier to dilute the initial sample thereby reducing the number of INPs in each aliquot. To account for
dilution, Eq. 1 can be rewritten as:
$$INP(T) = \frac{-ln(1-FF(T))DF}{V_a} \qquad (2)$$
where $DF$ is the dilution factor of the initial sample. However, in some cases dilution alone cannot be used to observe the total
number of $INP(T)$ due to the presence of impurities that act as INPs in the water used for dilution (Polen et al., 2018).
Therefore, it is necessary to use different bulk techniques that measure aliquots with volumes that span several orders of
magnitude, typically microliter to picoliter volumes (Harrison et al., 2018; Hill et al., 2014; Murray et al., 2010; Whale et al.,
77 2015).


Studies have investigated the concentrations of INPs in the atmosphere over the last 50 years and show that the concentration
in the atmosphere spans several orders of magnitude (Fletcher, 1962; Kanji et al., 2017; Petters and Wright, 2015; Welti et al.,
2018). Some of the original studies investigated the INP concentrations in melted hail and snow samples e.g. (Bigg, 1953;
Vali, 1971). Since then, studies have diversified to sampling INPs directly from the air (Boose et al., 2016b; Creamean et al.,
2013; DeMott et al., 2003; Lacher et al., 2017; Richardson et al., 2007; Welti et al., 2018), from precipitation (Christner et al.,
2008; Hill et al., 2014; Petters and Wright, 2015; Stopelli et al., 2015) and investigated potential types of INPs in the laboratory
from commercial and naturally occurring samples as well as field collected samples (Atkinson et al., 2013; Boose et al., 2016a;
Broadley et al., 2012; Felgitsch et al., 2018; Hill et al., 2014; Hiranuma et al., 2015, 2019; Kaufmann et al., 2016; Murray et
al., 2012; Pummer et al., 2012; Wex et al., 2015). Yet the atmospheric variability in INP concentrations remains unresolved
(Hoose and Möhler, 2012; Kanji et al., 2017; Petters and Wright, 2015; Welti et al., 2018). In order to further quantify the
variability of INP responsible for ice formation in MPCs and increase the fundamental understanding of ice nucleation, we
developed and characterized the DRoplet Ice Nuclei Counter Zurich (DRINCZ), a drop freezing instrument to investigate ice
nucleation at temperature conditions between -25 ˚C and 0 ˚C, representative for MPCs. DRINCZ complements and extends
the INP concentration measurement capabilities of the single particle and bulk methods employed at ETH Zürich e.g. (Kohn
et al., 2016; Lacher et al., 2017; Lüönd et al., 2010; Marcolli et al., 2007; Stetzer et al., 2008). Furthermore, the automation of
DRINCZ and its portable design allows for the acquisition of INP data in the field and laboratory, ultimately increasing the
attainable information about the global distribution of INPs.



## 2 Instrument Design

DRINCZ is based on the design of Stopelli et al. (2014) and Hill et al. (2014). It consists of a temperature controlled ethanol bath (Lauda ProLine RP 845, Lauda-Königshofen, Germany), a home-built LED light consisting of several LED light strips enclosed in an ethanol proof housing, a home-built 96-well tray holder and camera mount, a webcam (Microsoft Lifecam HD-3000) and a custom designed bath leveler, composed of a bath level sensor and valve (see Section 2.2) (Fig. 1a). The working principle is similar to that of Stopelli et al. (2014), in that a USB camera detects the light transmission through aliquots of sample. In DRINCZ, the aliquots are typically 50 µL and dispensed into a 96-well tray (732-2386, VWR, USA). To avoid contamination, the top of the 96-well tray is sealed with a transparent foil (Axgen, Platemax CyclerSeal Sealing Film, PCR-TS). The webcam is programmed to take a picture every 15 seconds, which corresponds to a picture taken approximately every 0.25 ˚C decrease when the bath is cooled at a rate of 1 ˚C min$^{-1}$. Moreover, both the picture frequency and cooling rate are adjustable. Upon freezing, the light transmission through an individual well decreases (red circled well in Fig. 1b) due to the polycrystallinity of the ice frozen in the wells.

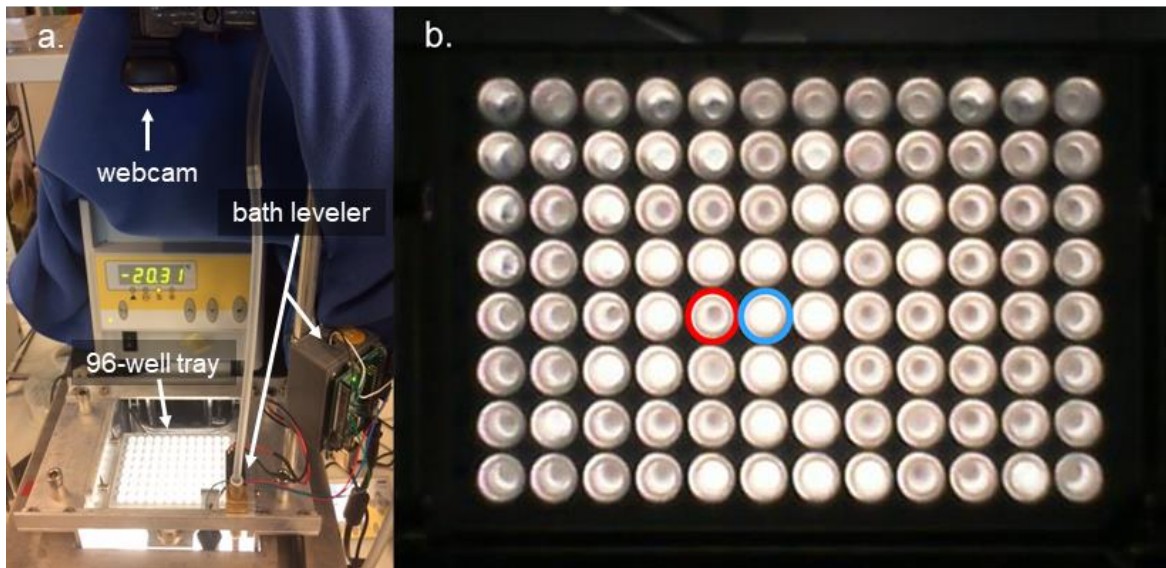

**Figure 1: (a) Picture of DRINCZ. (b) Change in light transmission through the wells during an experiment with an example of an unfrozen (blue circle) and frozen (red circle) well.**

The cooling cycle of the ethanol-based Lauda bath is controlled using LabVIEW® and the bath temperature is written to a text file that is then read in by MATLAB® during the analysis. In addition, MATLAB® is also used to take and save the pictures from the webcam. Both the LabVIEW® generated text file and pictures from the experiment are stored in the same folder for data handling. A suite of MATLAB® functions have been written to automatically analyze and store the data from each experiment, allowing for minimal user input (details of the code are provided in Appendix A) and rapid experiment throughput of approximately 30 minutes per experiment and 2 minutes to process the data for frozen fraction as a function of temperature.



## 2.1 Detection Method


The ice nucleation detection in DRINCZ is achieved by the attenuation of visible radiation due to a frozen well compared to
transmission through a supercooled well. The images are analyzed by first detecting the pixels that correspond to each well of
the 96-well tray and then calculating the change of the average well brightness during an experiment between one picture and
the next. The well detection method is described in the following subsection, followed by the technique used to detect well
freezing.

### 2.1.1 Circular Hough Transform for Well Detection


A fixed 96-well tray holder with an integrated webcam mount reduces variations in setting up the experiment. Nevertheless,
small changes in the location of the webcam due to mechanical shock during transport or testing, can produce misidentified
wells when algorithms rely on fixed well locations. Therefore, a freezing detection algorithm was developed to avoid errors
arising from small changes in the location of the wells.  To optimize contrast, the PCR tray holder was constructed out of
aluminum so that light transmission only occurs through the wells (see Fig. A1). The high contrast between the illuminated
wells and dark tray holder allows for the automatic detection of the wells using a Circular Hough Transform (CHT) (e.g.
Atherton and Kerbyson, 1999). The CHT first identifies pixels along regions of large brightness gradients such as the pixels
at the edge of the well. To determine the center of each well, the algorithm draws circles of varying diameter all centered at
the edge of the well and classifies the pixel intersecting the largest number of circles as the well center. The radius of the well
is then given as the radius of the circles that led to the highest number of intersections. The pixels within a well are then
identified as the ones encompassed by a circle drawn from a well center with the calculated radius as denoted by the red circles
in Fig. 2a. Since the CHT identifies the well center locations in random order, they must be sorted based on their x and y
coordinates using a pixel scale for spatial biases or refreezing results to be analyzed. The wells are sorted based on their center
locations using the following equation:
$$C_i = \frac{y_i}{D} L_x + x_i \qquad\qquad (3)$$
where $C_i$ is the value of the well center based on its pixel location in y and x coordinates, $y_i$ and $x_i$, respectively, with the
origin taken as the pixel in the upper left hand corner of the image. $L_x$ is the pixel number across the well array in the x





coordinate and $D$ is the diameter (pixel number) of the wells. All the $C_i$ values are then sorted to ensure that the wells are
identified based on their location independent of the experiment.

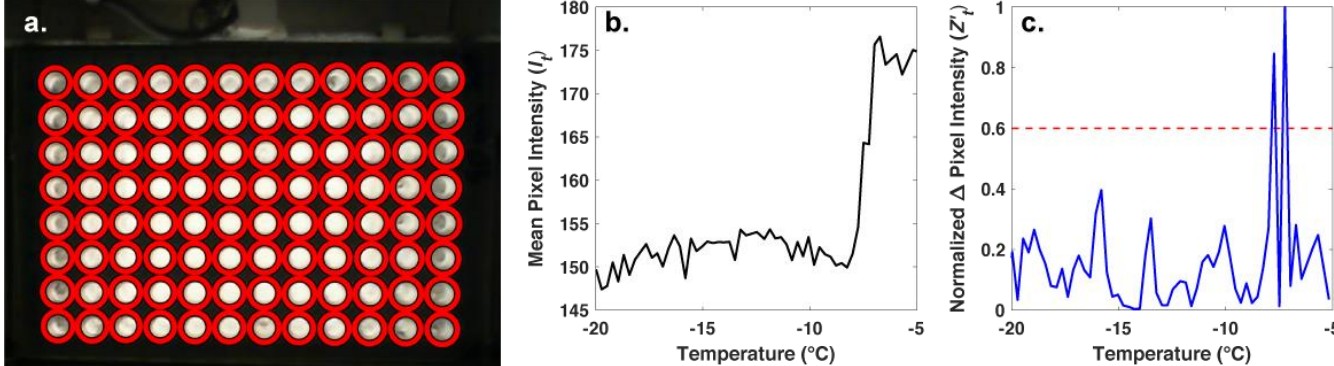

**Figure 2: (a) Automatic detection of the wells (red circles) using a CHT. (b) Light intensity or $I_t$ as a function of temperature as**
**observed by the webcam and (c) the normalized change in pixel intensity, $Z'_t$, between pictures taken during an experiment, as a**
**function of temperature. The dashed red line represents the 0.6 threshold required for a well to be classified as frozen.**

### 2.1.2 Freezing Detection

With the well locations identified, the intensity values of the pixels within each well are averaged for each image recorded
during an experiment ($I_t$). The change in $I_t$ between subsequent images is used to identify the image where freezing occurred
and the corresponding temperature (Fig. 2b). However, due to the slow freezing process which is limited by the latent heat
release, the light transmission of a well continuously changes until the water is completely frozen as can be seen as two large
peaks in Fig. 2c. To correctly identify the point in time when ice nucleation and not just freezing within the well occurs, the
maximum change in $I_t$ between subsequent images is normalized to 1 using the following procedure:
First, the Z-score ($Z_t$) of $I_t$ is taken to level out differences in illumination within the 96-well tray:
$$Z_t = \frac{I_t - \mu}{\sigma} \tag{4}$$
where $\mu$ and $\sigma$ are the mean and standard deviation of $I_t$ for all recorded images of a well, respectively. The absolute value of
the time derivative or the change in $Z_t$ between subsequent images ($dt$) is given as:
$$Z'_t = \left| \frac{Z_t}{dt} \right| \tag{5}$$
$Z'_t$ is then normalized to 1 by dividing by the maximum $Z'_t$ of the well. The normalization ensures that a fixed threshold for the
identification of ice nucleation can be used, independent of the absolute change in light transmission through a well during
initial freezing. Based on validation experiments, a threshold value of 0.6 $\left(\frac{z'_t}{\max(z'_t)} \geq 0.6\right)$ was found to be best for detecting
the initial freezing and to avoid assigning subsequent changes in transparency as a nucleation event due to slow freezing.

**2.2 Bath Leveler**

Due to the thermal contraction of the ethanol in the chilled bath between 0 and -30 ˚C, the ethanol level within the bath
decreases during an experiment, affecting the immersion level of the wells and thus the thermal contact. To keep the ethanol
level constant, a level sensor and solenoid valve are incorporated into the setup. The level sensor (Honeywell LLE 102101
liquid level sensor) detects when the ethanol falls below a fixed level relative to the wells and triggers the solenoid valve
(Kuhnke 64.025, 12 VDC valve) to open, allowing additional ethanol to flow into the bath. The level sensor and solenoid are
monitored and controlled using a 'sketch' written in Arduino (Arduino Uno Rev3 SMD). In order to minimize a possible
thermal gradient by adding warm ethanol to the bath, the ethanol is precooled to 0 ˚C using an ice water bath and then added
through a copper pipe that extends to the bottom of the bath. Thus, the bath leveler ensures that the wells remain in good
thermal contact due to a constant level of ethanol during experiments, while minimizing potential temperature fluctuations
within the bath. The resulting increased reproducibility of experiments due to the bath leveler is discussed in section 3.4.

**3 Validation**

The validation of the instrument is presented in four sections, with the first discussing the temperature calibration followed by
discussing the observed bias in freezing, the quantification of instrumental uncertainty and lastly, the improved reproducibility
of DRINCZ due to the addition of the bath leveller.

**3.1 Temperature Calibration**

The temperature reported as the freezing temperature is based on the ethanol bath temperature measured by the Lauda chiller
($T_{\text{lauda}}$). In order to correct for the difference between the temperatures of the sample in the wells ($T_{\text{well}}$) and $T_{\text{lauda}}$, a temperature
calibration was performed. The calibration was conducted by measuring the temperature (Type K thermocouple) within the
four corner wells and a center well of the 96-well tray (Fig. 3a). The wells were filled with ethanol instead of water to extend
the calibration across the entire experimental temperature range of DRINCZ without the interference of freezing. The
temperature bias between the wells and $T_{\text{lauda}}$ was measured every 1 ˚C while the bath was cooled at the typical ramp rate of
1 ˚C min$^{-1}$. The calibration was performed three times for each well (Fig 3b). Not surprisingly, we found that the ethanol
temperature in the bath was consistently lower than the temperature in the five calibration wells and the difference between
bath and well temperature increased linearly as the bath temperature decreased. Based on these results the linear function
$T_{corr} = 0.917 * T_{lauda} + 1.3$, with $T_{lauda}$ in ˚C (black line in Fig. 3b) was derived to correct the well temperature, with a
maximum standard deviation of ±0.6 ˚C.

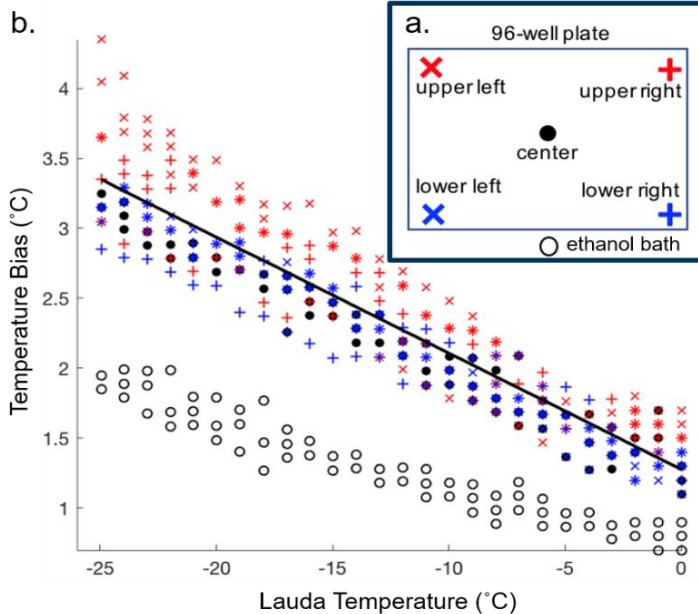

**Figure 3: (a) Locations of the type-K thermocouples tested during the temperature calibration. Additionally, the temperature difference between the Lauda temperature and the ethanol bath was measured at the indicated location (black open circle). (b) The temperature bias between the wells and ethanol bath is displayed versus the Lauda bath temperature. The linear temperature correction is shown in black.**

**3.2 Freezing Bias across the 96-well Tray**

The temperature calibration discussed above revealed potential variations in the well temperatures between the corner and the center wells. We thus quantified the bias for individual wells, but conclude that it is within the instrument experimental error as discussed below. To do so, 20 pure water (Molecular Biology Reagent, W4502 SigmaAldrich; hereafter referred to as SA water) experiments were analyzed. SA water was chosen for this analysis due to its homogeneity and low freezing temperature, where the observed spread in well temperature was maximized. For each well the median freezing temperature (or temperature when frozen fraction ($FF$) = 50 %) ($\widetilde{w}_i$) was compared to the median freezing temperature of the 4 corner wells ($\widetilde{w}_{4ref}$) used for the temperature calibration (see Fig. 3a). The difference between $\widetilde{w}_{4ref}$ and $\widetilde{w}_i$ ($\widetilde{w}_{4ref} - \widetilde{w}_i$) is shown in Fig. 4a. The red (blue) shading indicates a warm (cold) bias and signifies that the solution in these wells are exposed to warmer (colder) temperatures than the average of the four reference wells. The higher concentration of red shades in the middle of the tray suggests that the center of the tray is exposed to as much as 1.5 ˚C warmer ethanol flow than the tray periphery. Indeed, the ethanol circulates clockwise in the Lauda chiller and thus the freezing appears to track the flow (arrows in Fig. 4). Thus, the ethanol circulation explains the observed bias. The same analysis procedure was applied to the same 20 samples separated by user (12 and 8 experiments) and a similar bias was observed (see Appendix Fig. A2). Therefore, the reported bias is instrumental, reproducible and any potential user bias can be excluded. The bias was found to be statistically significant at the 95% confidence interval for 30% of the wells and resulted in an overall bias of 0.23 ˚C (see Fig 4b and Appendix A.). As such,

a well by well bias correction was developed and tested as described in Appendix A. Although the bias correction performed
as expected, the bias of 0.23 ˚C falls within the instrumental uncertainty as discussed in Section 3.3 and is therefore not applied
to DRINCZ measurements by default. Nevertheless, the potential benefits and impacts of a bias correction is discussed in the
following section.

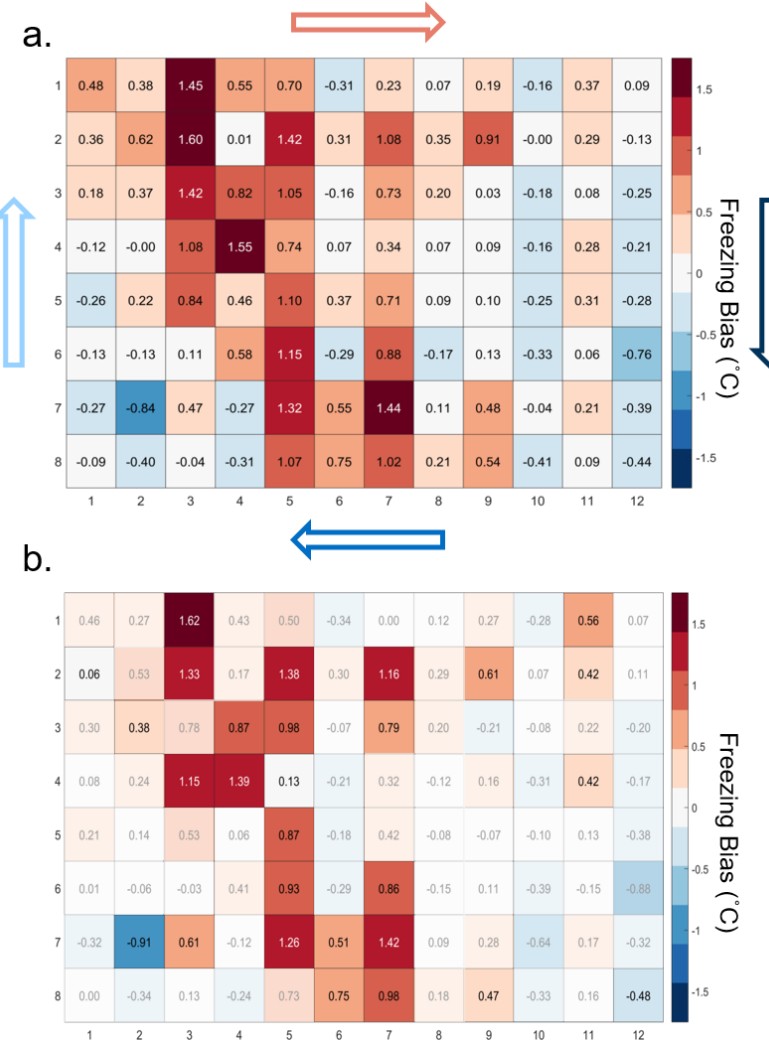


**Figure 4: (a) Bias in the freezing of SA water ($\tilde{w}_{4ref} - \tilde{w}_i$ in ˚C) based on the median value of each well over 20 experiments relative to the median temperature of freezing for the 4 corner wells used during the temperature calibration. A positive (negative) bias indicates that the wells experience a warmer (colder) temperature than the four corner wells used for temperature calibration and therefore freeze at lower (higher) temperatures than reported. The arrows represent the ethanol circulation in the chiller and the color represents the temperature trend of the ethanol as it circulates in the bath with dark blue being the coldest and red the warmest. (b) Mean freezing bias of SA water between the four reference wells and each well ($\bar{w}_{4ref} - \bar{w}_i$). Positive (negative) values indicate, as denoted by shades of red (blue), wells that systematically freeze at colder (warmer) temperatures and therefore experience warmer (colder) temperatures than reported. Statistically insignificant biases as determined by a Welch's $t$-test (see Eq. A1) are depicted as greyed out.**

### 3.2.2 Impact of Bias Correction on Frozen Fraction


By accounting for the bias in freezing temperature across the 96-well tray by first applying the temperature calibration and
then the bias correction such that corrected well value ($\bar{\bar{w}}_i$) becomes:
$$\bar{\bar{w}}_i = \bar{w}_i + (\bar{w}_{4ref} - \bar{w}_i),\qquad(6)$$
the slope of the *FF* curves steepens and becomes smoother, which is expected as the observed freezing temperatures become
more constrained (see Fig. 5). Although the median freezing temperature with and without the bias correction only changes by
0.2 ˚C (consistent with the correction of the mean bias of 0.23 ˚C found above), the narrowing of the freezing temperature
distribution is significant at the 95% significance level (Welch's *t*-test, see Eq. A1). This result shows that by using the spatial
dependent freezing information of a well from optically based drop freezing instruments like DRINCZ, temperature can be
better constrained. Such a bias correction should also be applicable to freezing methods that use block based cooling, where
gradients across the block may exist (Beall et al., 2017; Harrison et al., 2018).

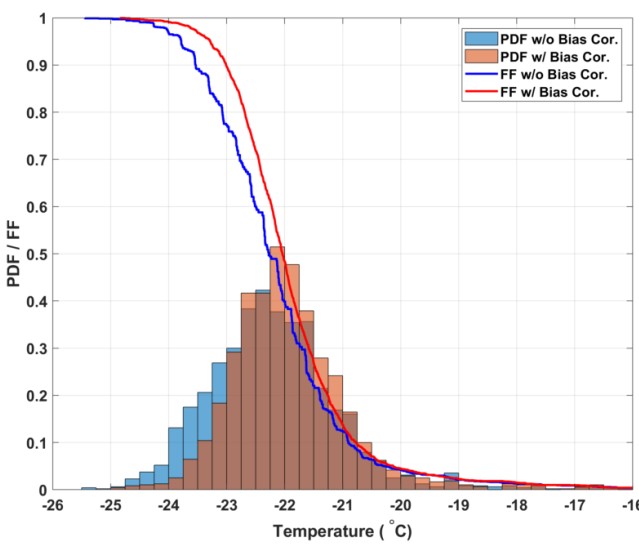


**Figure 5: Histograms representing the probability distribution functions for freezing temperatures of the 20 SA water experiments**
**without (blue bars) and with the bias correction (red bars). The calculated cumulative distribution functions, or frozen fraction**
**curves without and with the bias correction are represented as the blue and red lines, respectively.**

### 3.3 Instrument Uncertainty


The instrumental uncertainty for DRINCZ is assessed by using the standard deviation in freezing temperature of the SA water
experiments in combination with the error in the temperature of the wells established during the temperature calibration. The
standard deviation of the freezing temperature of the SA water is dependent on *FF*, with a minimum at 50% *FF* (Fig. 6a). This
dependence is expected as the 50% *FF* corresponds to the most likely temperature for the SA water to freeze and therefore,
should show the least variability across the 20 experiments used in the analysis. The standard deviation at each *FF* is the





uncertainty due to the instrument as well as the variability in the freezing temperature of the SA water and represents the upper
limit of the instrumental uncertainty. Given the contribution to the uncertainty due to the variability of the freezing temperature
of the SA water, the standard deviation at $FF = 50$ % can be used as the upper limit of the instrumental uncertainty across the
entire $FF$ range. Incorporating a bias correction results in a negligible average difference in the standard deviation (as shown
by dashed lines in Fig. 6a). Thus, the upper limit of the instrumental precision is ± 0.3 ˚C (the mean of the standard deviation
of freezing temperature over the entire freezing spectrum).

Although the instrumental precision indicates that DRINCZ is very reproducible (± 0.3 ˚C), the accuracy in the reported
temperature must be accounted for. Based on the temperature calibration, the standard deviation of the well temperatures is
temperature dependent. At the coldest temperatures of the freezing range of the SA water (~ -25 ˚C), the standard deviation of
the well temperatures is largest, likely due to the increased gradient between the bath and air temperature and therefore, the
importance of the ethanol circulation through the bath is increased. To account for this temperature dependence, the maximum
standard deviation of ± 0.6 ˚C from the temperature calibration, corresponding to the lowest observable freezing temperature
in DRINCZ (freezing temperature of SA water) is used. Therefore, when accounting for both the precision of the measurements
and the accuracy of the temperature, the overall uncertainty of the reported freezing temperature of a well in DRINCZ is
± 0.9 ˚C. This value is comparable to other recently developed drop freezing techniques, which report uncertainties ranging
between ± 0.9 ˚C (Harrison et al., 2018) and ± 2.2 ˚C (Beall et al., 2017).

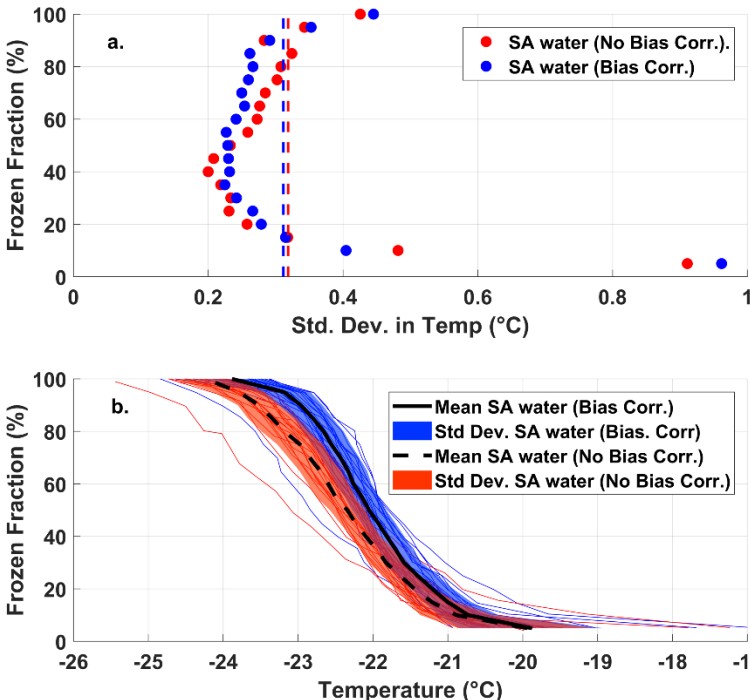


**Figure 6: (a) FF and the corresponding standard deviation of the freezing temperatures from the 20 SA experiments with and**
**without the bias correction shown as blue and red dots, respectively. The red and blue dashed lines represent the standard deviations**



**in temperature averaged over all FF values without and with the bias correction, respectively. (b)The FF of the 20 SA water**
**experiments as a function of temperature with and without the bias correction (thin blue and red lines, respectively). The color fill**
**represents the standard deviations of the SA water from the mean freezing temperature with (solid black line) and without (the**
**dashed black line) the bias correction.**

**3.4 Importance of the Bath Leveler**
To assess the impact of the decreasing ethanol level on experiments in DRINCZ, 32 experiments with SA water without a bath
leveler were compared to the 20 SA water with a bath level sensor, the same 20 SA water discussed in the previous section.
Figure 7a shows that the bath sensor reduces the spread in freezing temperatures observed. The decrease in the 50 % *FF*
temperature without the bath leveler is due to a larger gradient between the aliquot and the bath temperatures, thus the well is
warmer than expected, requiring further cooling to observe freezing. The additional cooling in combination with the variable
starting level of the ethanol relative to the wells in the cases of no bath leveler is responsible for the longer freezing tail of the
*FF* curve (blue line) at higher *FF*s. Without the bath leveler, the initial height of ethanol relative to the wells is user dependent
and not reproducible, leading to both the higher and lower observed freezing temperatures.

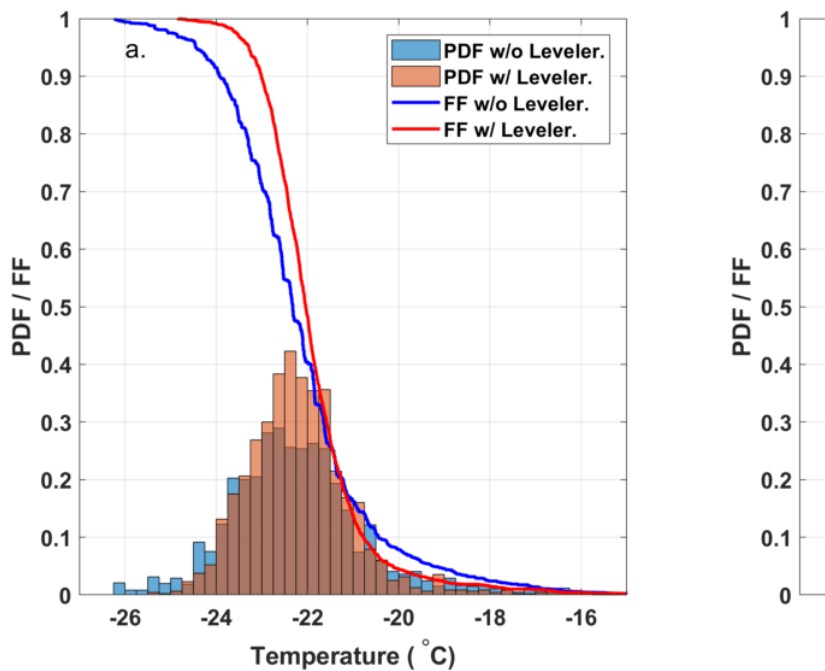
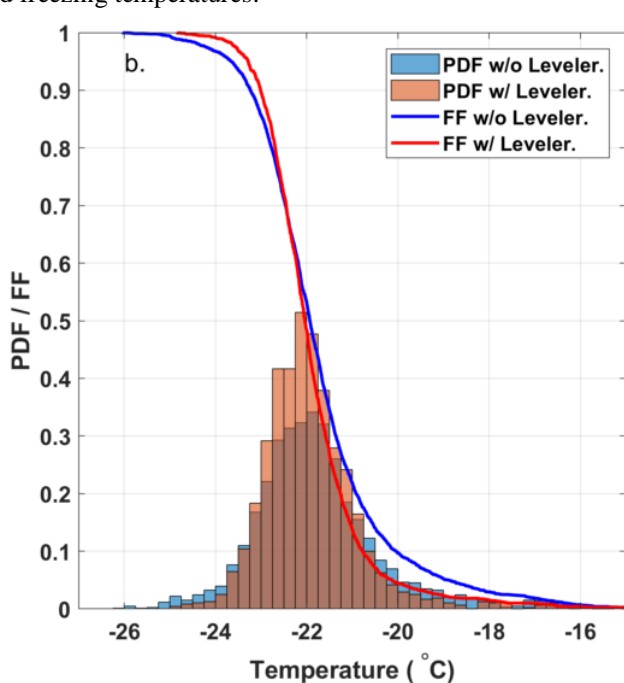


**Figure 7: (a) Comparison of the freezing temperature of SA water without (32 experiments, blue) and with (20 experiments, red) the**
**bath leveler. The histograms are normalized to represent the PDF of the freezing temperatures and the lines represent the mean FF**
**curves of the SA water experiments. (b) Shows the same as panel a, except that a bias correction is applied to both sets of experiments.**
Although the median freezing temperature (*FF*=50 %) only decreased by 0.25 °C without the bath leveler, the freezing curves
steepen when the bath leveler is incorporated in DRINCZ, leading to a decrease of the standard deviation from ± 1.4 to ± 1.0 °C
over the entire *FF* range. A bias correction applied following the procedure in Section 3.2 reduces the issues associated with



a variable bath level as seen by the similar *FF* curves and histograms normalized using the probability density function (PDF)
estimate in Fig. 7b for experiments with and without the bath leveler. The difference in mean freezing temperatures decrease
to 0.05 ˚C at *FF*=50 % and the standard deviation of the SA water freezing temperature without the leveler decreases from ±
1.4 to ± 1.2 ˚C over the entire *FF* range. This decrease is expected as the bias correction is designed to reduce the spread in
freezing temperatures within the 96 aliquots. Although the bias correction reduces the need for a bath leveler in DRINCZ, the
bias is instrument dependent and may be less pronounced in other drop freezing setups. Therefore, we recommend the use of
a bath leveler in any bath-based drop freezing devices.

## 4 Freezing Experiments

To verify the performance of DRINCZ in the context of other published drop freezing techniques, we use the SA water
experiments to characterize the instrumental background (Section 4.1) and perform freezing experiments with NX-illite
suspensions (Section 4.2). To demonstrate applicability of the instrument to analysis of field samples, the evolution of the ice
nucleating ability of atmospheric aerosol particles collected in snow samples at the Sonnblick Observatory in the Hohe Tauern
region of Austria during a mid-latitude storm system is assessed in Section 4.3. Lastly, some uncertainties associated with
measuring INP in snow samples (Section 4.4) and further validation of DRINCZ through dilutions are discussed (Section 4.5).

### 4.1 Background of DRINCZ

The background freezing due to the experimental technique and the SA water used to suspend and dilute samples must be
known to discriminate freezing events due to the sample from freezing events due to the water used. The 20 SA water
experiments are therefore used to assess the instrument background freezing. It is important to note that in cases where solvents
other than SA water are used or where contamination from a sampling technique (e.g. snow collection or impinger
measurements) is possible, a different background calculation must be used to accurately assess the freezing ability of a sample.
The background of DRINCZ when used with SA water, is calculated by fitting the 20 SA water experiments with a five
parameter Boltzmann fit. The five parameter version was chosen to account for asymmetry (Spiess et al., 2008) in the freezing
of the SA water but due to the minimum and maximum values of FF given as 0 and 1, respectively, the fit reduces to three
parameters and takes the form:
$$FF_{BGfit}\left(T_{frzBG}, a, b, c\right) = \frac{1}{\left(1 + e^{a\left(T_{frzBG} - b\right)}\right)^c},\tag{8}$$
where $FF_{BGfit}$ is the fitted *FF* of the SA water as a function of the observed freezing temperatures of the SA water, $T_{frzBG}$,
and the fitting parameters, $a$, $b$, $c$ represent the slope of the fit ($a$ = 1.9651), the inflection point ($b$ = -22.7134) and the
asymmetry factor ($c$ = 0.6160), respectively. The value of 1 in the numerator represents the maximum *FF*. The fit and
associated coefficients (including 95% confidence range and $r^2$) are shown in Table 1 and Fig. 8 respectively.




**Table 1: Coefficients for the three parameter Boltzmann fit of the SA water freezing background and 95th percentile confidence interval bound values.**

|          | $a$     | $b$       | $c$    | $r^2$ |
|----------|---------|-----------|--------|-------|
| Best     | 1.9651  | -22.7134  | 0.6160 | 0.97  |
| -95th %  | 1.7254  | -22.8955  | 0.4683 | N/A   |
| +95th %  | 2.2049  | -22.5312  | 0.7637 | N/A   |


The fitted freezing background is used to correct for the contribution of SA water to the observed freezing of a sample. To account for the presence of multiple ice nucleating particles coexisting in a single well, the background is removed by subtracting the differential nucleus concentration of the background from that of the sample (Vali, 1971, 2019). The differential nucleus concentration ($k(T)$) is initially defined in Vali (1971) as:

$$k(T) = -\frac{1}{V_a \Delta T} \cdot ln\left(1 - \frac{\Delta N}{N(T)}\right),$$ (9)

where $N(T)$ is the number of unfrozen aliquots at the beginning of a temperature step while $\Delta N$ is the number of aliquots that freeze during the temperature step (between pictures) or $\Delta T$.

The background corrected differential nucleus concentration ($k_{corr}(T)$) is obtained by:

$$k_{corr}(T) = k_{sam}(T) - k_{bg}(T),$$ (10)

where $k_{sam}(T)$ and $k_{bg}(T)$ are the sample and background differential nucleus concentration, respectively. The background corrected $FF_{cor}(T)$ is then achieved by inverting Eq. 9 and taking the cumulative sum of $k_{corr}(T)$:

$$FF_{cor}(T) = 1 - exp(-\sum[k_{corr}(T) \cdot \Delta T] \cdot V_a),$$ (11)

An example of the impact of the background correction on the *FF* of the diluted snow sample collected on Nov 30th 2017 (discussed in section 4.3) is shown in Fig. 8.





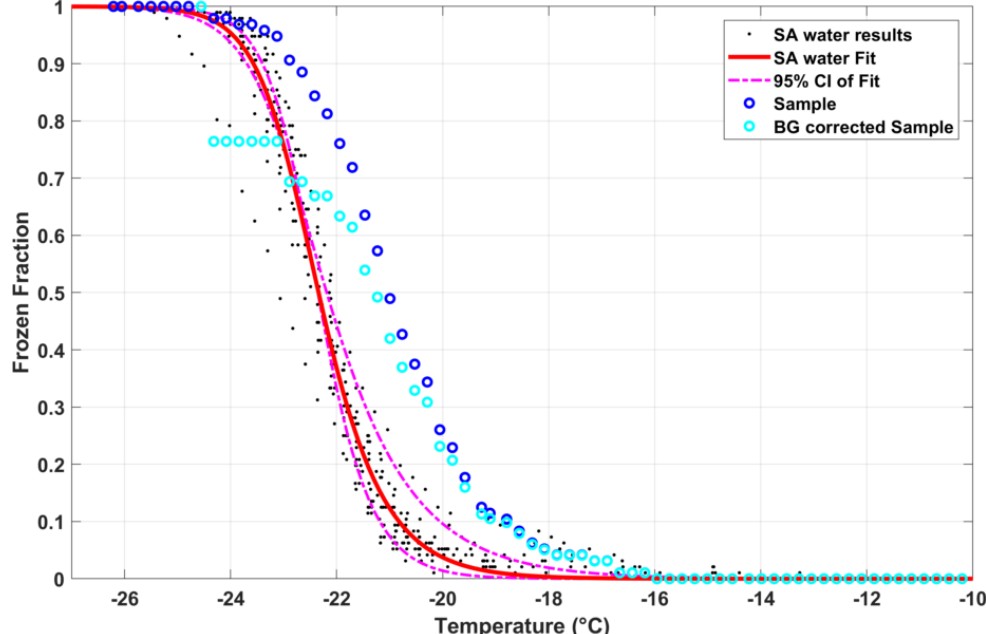

**Figure 8: SA water data (black dots) and corresponding fit (red line, Eq. 8) including the 95th percentile confidence interval (dashed-dot magenta lines). The blue circles represents the diluted snow sample collected on Nov 30th 2017 which is then corrected for the contributions of freezing from the SA water using the background correction ($FF_{cor}(T)$ as described in Eq. 11; cyan circles).**

### 4.2 Comparison of DRINCZ to other immersion freezing techniques

To validate the performance of DRINCZ, we use different wt. % NX-illite suspensions to compare the results from DRINCZ to those summarized in Hiranuma et al. (2015), Beall et al. (2017) and Harrison et al. (2018). In the atmosphere, illite constitutes up to ~40 % of the transported dust fraction (Broadley et al., 2012; Murray et al., 2012), making it an excellent surrogate for atmospherically relevant dust. Three mass concentrations of NX-illite (0.01, 0.05, and 0.1 wt. %) were investigated with DRINCZ (see Fig. A4 for $FF$ curves) and then normalized to the number of active sites per BET-derived surface area ($n_{sBET}$) using a variation of Eq. 2 as follows:

$$n_{sBET} = -\frac{\ln(1-FF)}{V_a * SA_{BET} * C_{NX}},$$ (12)

where $SA_{BET}$ is the BET surface area of the particles used (NX-illite) and $C_{NX}$ is the mass concentration of NX-illite in an experiment.

The $n_{sBET}$ of NX-illite calculated using Eq. 12 from the measurements made with DRINCZ falls within the results from Hiranuma et al. (2015), Beall et al. (2017) and Harrison et al. (2018) (Fig. 9). In theory, $n_{sBET}$ should be insensitive to concentration as the number of ice nucleating sites is normalized to the total surface area. Indeed, the differing weight percent samples overlap (Fig. 9). Furthermore, the lower-weight-percentage samples extend the observable $n_{sBET}$ to higher values and



colder temperatures, as expected. Similar to the observations of Harrison et al. (2018), a few of the data points from the
0.01 wt. % solution appear as outliers at the warmest temperatures. This is likely due to an uneven distribution of the active
sites in each aliquot and may be the result of diluting a single stock solution rather than producing individual weight percent
solutions (Harrison et al., 2018). Thus a spread equivalent to or less than the spread in the concentrations, up to an order of
magnitude in this case, can be expected. Furthermore, when accounting for the ± 0.9 ˚C uncertainty, depicted by the horizontal
error bars, the results between concentrations become more similar and fall within the same range as the measurements of
Beall et al, (2017) who used similar concentrations of NX-illite. The overlap between the $n_{sBET}$ measured with DRINCZ and
the NX-illite parameterization (Hiranuma et al., 2015) indicate that DRINCZ is capable of accurately measuring the
concentration of INPs and their active sites in the immersion freezing mode (Fig. 9).

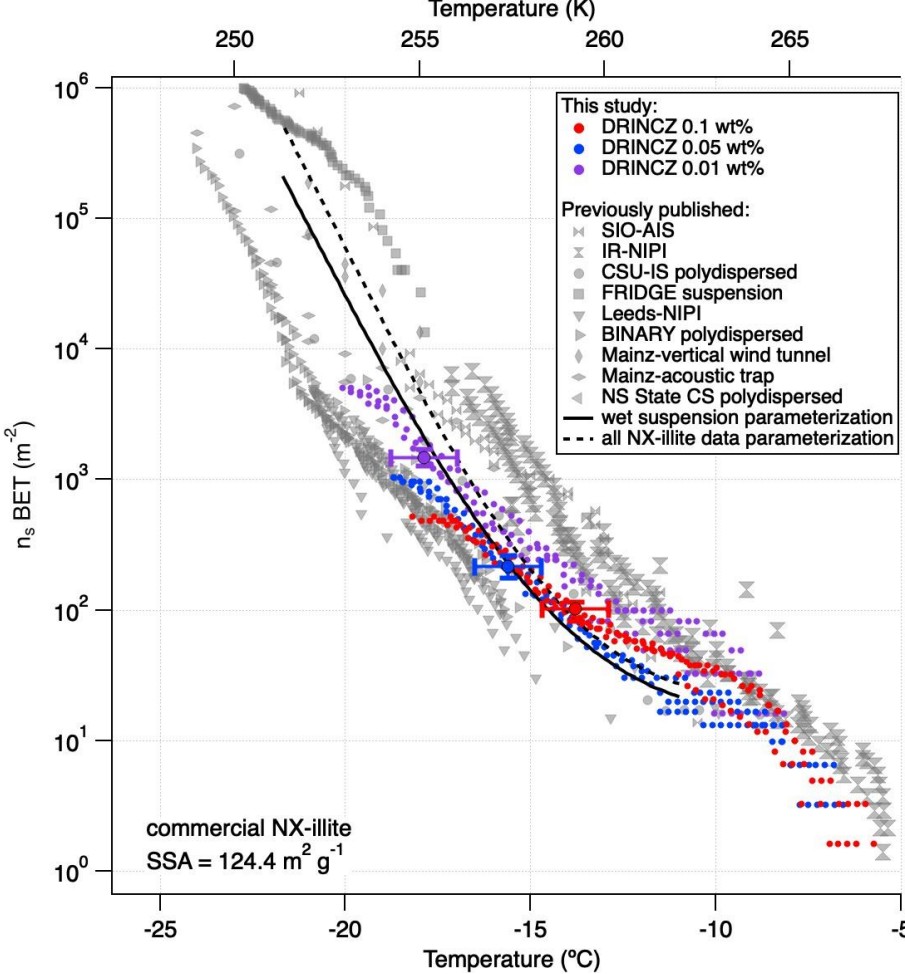

**Figure 9:** $n_{sBET}$ of three concentrations of NX-illite, $10^{-3}$ g ml$^{-1}$ (red dots), $5\times10^{-4}$ g ml$^{-1}$ (blue dots) and $10^{-4}$ g ml$^{-1}$ (purple dots), are measured as triplicates by DRINCZ and reported as a function of temperature. An example of the temperature uncertainty and the uncertainty due to the background correction are depicted for each weight percent as horizontal and vertical error bars, respectively. Literature values from Hiranuma et al, (2015), Beall et al, (2017) and Harrison et al, (2018) are shown for comparison. $n_{sBET}$ was calculated using a BET surface area of 124.4 m$^2$g$^{-1}$ (Hiranuma et al., 2015).



372

### 4.3 Ice Nucleating Particle Concentrations in Snow Samples from a Mountaintop Observatory in Austria

In order to demonstrate the performance of DRINCZ, snow samples collected between the 27th and 30th of November 2017 at the Sonnblick Observatory (SBO) were analyzed. The SBO is located at 3106 m on the summit of Mt. Sonnblick in the Hohe Tauern Region of Austria and has previously been used for cloud microphysical measurements (e.g. Beck et al., 2018; Puxbaum and Tscherwenka, 1998). Freshly fallen snow was collected from a wind-sheltered area where the snow could not drift. A stainless steel shovel (Roth) was conditioned with snow by turning (10 times) in the surface snow next to the sampling site prior to sampling. The snow was then sampled into sterile NascoWhirlPaks (Roth) and then melted at room temperature (20 ˚C), immediately after which aliquots of snow-meltwater were filled into sterile centrifugation tubes (15 ml, Falcon tubes) and stored at -20 ˚C. The samples were stored frozen until processed with DRINCZ to avoid any bacterial growth or changes due to liquid storage (Stopelli et al., 2014). The snowfall collected at SBO occurred during two snowfall events. The first event began on the 25th and ended overnight on the 26th (early hours of the 27th) while the second event (28th -30th) was associated with an intensifying upper level trough, a developing surface cyclone, a strong cold front and an associated secondary low (see Fig. A5 and A6).

The frozen fractions of five different snow samples were determined using DRINCZ and the cumulative concentration of active sites (or *INP(T)*, see Eq. 1) were normalized to per L of meltwater ($n_{mw}$) (Fig. 10). Overall, the $n_{mw}$ of the snow samples fall within the range of previously reported values for precipitation samples (Petters and Wright, 2015) except for the November 30th sample. Within these samples, we identify (1) a particularly active snow sample (Nov 28th), (2) samples having intermediate IN activity (Nov 27, 29), and (3) a least active sample (Nov 30th). We attempt to compare these snow samples based on their air mass origin.

The snowfall sampled on the 28th had the highest $n_{mw}$ of all collected samples (Fig. 10). The meteorological conditions and a comparison of back trajectories indicate that the air mass was associated with the warm sector of a synoptic system (Fig. A7) that originated from North America and the North Atlantic that then crossed France and Switzerland, before arriving at SBO (Fig. A8). In contrast, the arctic air mass responsible for the snowfall sampled on the 27th originated over Svalbard before crossing Iceland, the British Isles, Northern France and Germany (Fig. A8).

Even though the local conditions at SBO did not change significantly between the 28th and 29th, a decrease in $n_{mw}$ was observed relative to the 28th and $n_{mw}$ gradually decreased between the first and second sample on the 29th (Fig. 10). The back trajectories show that the origin of the air mass changed from North America and the North Atlantic on the 28th to exclusively originating over the North Atlantic on the 29th (Fig. A8). Additionally some of the back trajectories on the 29th show an increased interaction with the boundary layer over Europe (Fig. A8). Nevertheless, the decrease in $n_{mw}$ suggests that if boundary layer



aerosols from parts of Europe did reach the precipitating clouds at the SBO, they are less efficient INPs than the marine aerosols
(Lacher et al., 2017, 2018) associated with the samples on the 27[th] and 28[th].

Finally, the lowest $n_{mw}$ observed were from meltwater collected on the 30[th]. The cold frontal passage and associated cold air
advection caused the temperature to drop by 6 ˚C by noon on the 30[th] (Fig. A7) and the $n_{mw}$ in the associated snowfall decreased
substantially, exceeding the lower limit of previously reported $n_{mw}$ values (Petters and Wright, 2015, Fig. 10). The decrease in
$n_{mw}$, however, cannot be explained solely on the origin of the air mass as the arctic air mass on the 27[th] also crossed similar
parts of the UK or had significant interaction with the marine boundary layer. Nevertheless, the concentration of INPs in the
sea surface microlayer is variable and the efficiency of emitting marine INP from the surface is wind speed dependent (DeMott
et al., 2016; Irish et al., 2017; McCluskey et al., 2018; Wilson et al., 2015). Therefore, even though the trajectories on the 27[th]
and 30[th] interacted with the marine boundary layer, they may contain different concentrations of INPs, yielding the observed
differences in $n_{mw}$. In addition to air mass origin, it has been shown that precipitation efficiently removes INP and thus
influences $n_{mw}$ (Stopelli et al., 2015). Indeed, the most upstream precipitation (see Fig. A8) corresponds to the sample collected
on the 30[th], which has the lowest $n_{mw}$. Therefore, the most efficient INPs could have been removed in the upstream
precipitation, contributing to the observed decrease in $n_{mw}$.

The differences in $n_{mw}$ could not be rectified by a single metric in this study but rather a combination of factors likely led to
the observed variability. In particular, as the warm sector of the cyclone approached the sampling site (28[th]), $n_{mw}$ increased.
Conversely, after cold frontal passage (30[th]) the $n_{mw}$ decreased. Back trajectories indicate that the air mass source region and
the amount of upstream precipitation differed between the two sectors of the cyclone. This result is consistent with previous
studies that suggest that air mass origin (e.g. Ault et al., 2011; Creamean et al., 2013; Field et al., 2006; Lacher et al., 2017,
2018) and upstream precipitation (Stopelli et al., 2015) influences the INP concentration. Furthermore, the dependence on the
long range air mass history to the observed variability in $n_{mw}$ suggests that local sources are not responsible for the observed
INPs.

### 4.4 Limitations of snow meltwater sample comparisons

One limitation when comparing snow samples collected at different times and locations is the unknown number of aerosols,
INPs and ice crystals that contributed to the collected meltwater. Since $n_{mw}$ depends on the number and mass of the ice crystals
within a snow sample, the melt water volume or density of each snowflake influences $n_{mw}$. For example, snow to liquid ratios,
which can be used as a proxy for snow flake density and melt water equivalent, can vary between 5 to 1 in heavy wet snow
and 100 to 1 in powdery snow (Roebber et al., 2003). However, even when considering this variability in the required amount
of snow to produce the same volume of ice crystal melt water, $n_{mw}$ would only differ by a factor of 20. As can be seen in
Fig. 10, $n_{mw}$ varies by two orders of magnitude or more between the 28[th] and the 30[th] of November and the difference is
therefore robust. Additionally, heavy wet snow has been found to occur in the warm core of a synoptic system while lighter,





more powdery snow was found in the air mass after cold frontal passage, where air temperatures are colder (Roebber et al.,
2003). As the $n_{mw}$ on the 28[th] was collected in the warm sector and the sample on the 30[th] was post cold front, differences in
snow density may lead to an underestimation in the difference between the $n_{mw}$ of these two samples. Therefore, we recommend
that future studies also consider the snow water equivalent when comparing the $n_{mw}$ as this could influence $n_{mw}$ by a factor of
20 or more.

Another uncertainty with using precipitation samples for analyzing INP concentrations is associated with aerosol scavenging
and chemical ageing (e.g. (Petters and Wright, 2015). As previously mentioned, the samples were stored frozen to avoid any
decrease in ice nucleating ability associated with storage (Stopelli et al., 2014) and therefore degradation is likely not an issue
in this study (Wex et al., 2019). The ability of a falling ice crystal to scavenge aerosols or rime cloud droplets depends on the
ice crystal habit, size, and the difference between the fall velocity of the crystal and the interstitial aerosol or cloud droplets.
With the exception of interstitial aerosol concentration which has been shown to influence $n_{mw}$ by a factor of 2 (Petters and
Wright, 2015), these factors are all important when estimating snow density and thus make it difficult to disentangle their
effects on $n_{mw}$. Therefore, there is value in future studies of INP in MPCs to investigate the INP concentrations in cloud water,
interstitial aerosols and snow samples.
**4.5 Ice Nucleating Particle Concentrations in Diluted Snow Samples**
In order to extend the reported temperature range of DRINCZ, the snow samples were also diluted by a factor of 10 with SA
water (see Eq. 2). The dilutions (open symbols) overlay the pure samples except at the warmest temperatures where, as
previously mentioned, a single freezing event can lead to an increase in $n_{mw}$ of an order of magnitude relative to the undiluted
sample. This effect is especially evident on the 27[th] when the first few wells of the diluted sample (open blue circles) froze at
the same or higher temperatures than the undiluted sample (filled blue circles) and led to an increase in $n_{mw}$ of up to an order
of magnitude. However, this issue has been previously observed when diluting from stock solutions (Harrison et al., 2018)
which is similar to diluting a snow water sample. Therefore, the dilutions further validate DRINCZ as an INP measurement
technique.



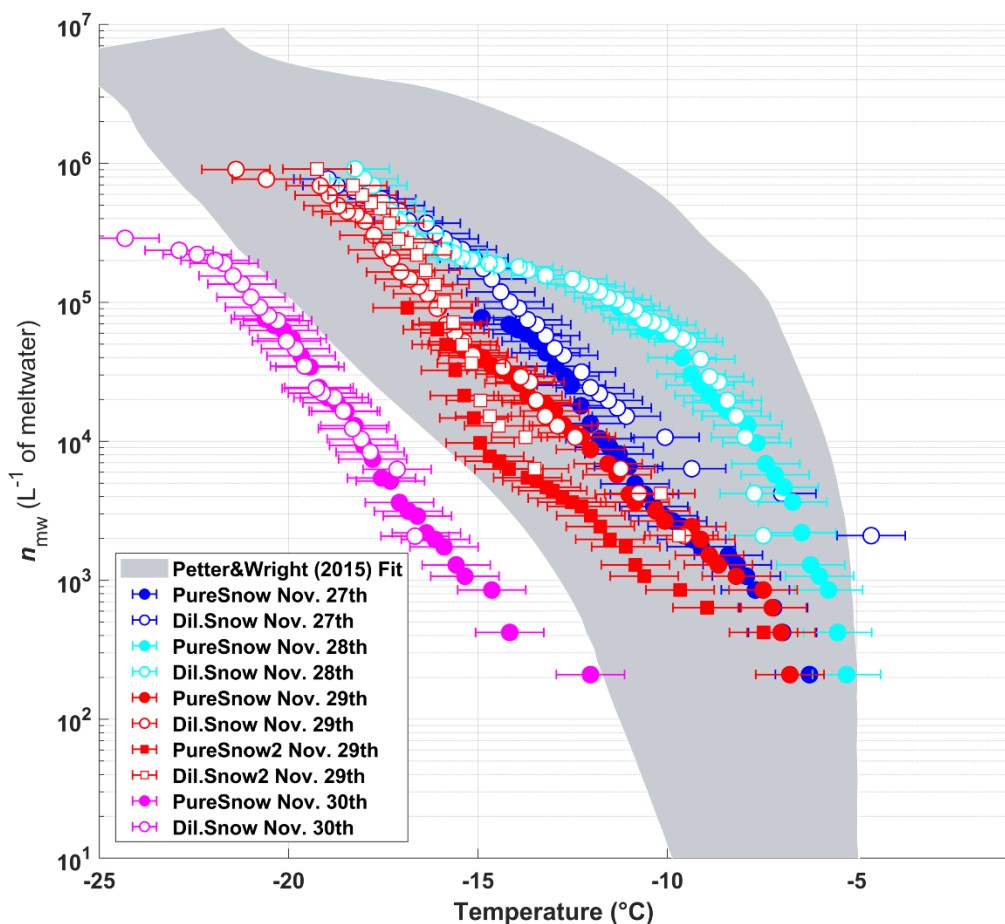

**Figure 10: The cumulative number of active sites per L of meltwater ($n_{mw}$) of snow for undiluted snow (filled) and of snow samples diluted by a factor of 10 (white-filled symbols) as a function of temperature. The colors represent the different sampling days. On the 29th of Nov. two samples were taken and the second sample of the day is indicated by square symbols. The shaded area represents the previously reported $n_{mw}$ from precipitation events as described in Petters and Wright (2015). The error bars represent the instrumental temperature uncertainty of ± 0.9 ˚C.**

## 5 Conclusions

We describe and characterize DRINCZ as a newly developed drop freezing instrument for quantifying the ability of aerosols to act as ice nucleating particles in the immersion freezing mode. The instrument uncertainty is ± 0.9 ˚C, similar to previously published drop freezing techniques. We show that thermal contraction of ethanol as a coolant used in bath-based drop freezing techniques increases temperature variations within the sample. This issue can be corrected by incorporating a bath leveler which ensures the coolant level in the bath remains constant during an experiment. Typical drop freezing methods report temperature measured in the corner wells of a 96-well tray, at the edge of a cooling block or within the block itself (Beall et al., 2017; Hill et al., 2014; Stopelli et al., 2014). Here we show that by making use of the freezing sequence of pure water aliquots, the spatial pattern of temperature bias in the 96-well tray can be assessed. Although variations are within the



instrumental uncertainty of DRINCZ and are not used for DRINCZ data analysis, we present our detailed analysis of this
potential bias and draw attention to this issue for other drop freezing techniques. The calculated bias correction increases the
precision of drop freezing setups, and is an alternative to computationally expensive heat transfer simulations (Beall et al.,
2017). Validation experiments conducted with NX-illite showed good agreement with data reported in the literature for this
INP standard.
We exemplify the use of DRINCZ by measuring the concentration of INP in snow samples collected at the Sonnblick
Observatory in Austria. The observed INP concentrations are within previously reported values as summarized in Petters and
Wright, (2015) for the same temperature range as investigated here (-22 to 0 ˚C). Differences in INP concentration can be
explained by differing sectors of a mid-latitude cyclone. As the warm sector of the cyclone approached the sampling site, the
INP concentration increased while after the cold front passed the INP concentration decreased. Back trajectories indicate that
the air mass source region and the amount of upstream precipitation differed between the two sectors of the cyclone. This
result is consistent with previous studies that suggest that air mass origin (e.g. (Ault et al., 2011; Creamean et al., 2013; Field
et al., 2006; Lacher et al., 2017) and upstream precipitation (Stopelli et al., 2015) influence the INP concentration. This suggests
that INP in precipitation samples are likely transported from specific source regions rather than originate from local sources.
Thus identifying the specific sources responsible for INP and their transport pathways are essential for accurately modelling
the ice phase in clouds and ultimately, climate.

## Author Contributions

DRINCZ was developed and designed by R.O.D with the assistance of M.C.C, M.R., L.S.B, K.P.B and N.B.D. The SA water
experiments were conducted by M.C.C, K.P.B., L.S.B, V.W, J.W, S.B, and R.O.D. The temperature calibration and NX-illite
experiments were conducted and analyzed by K.P.B and N.B.D. The snow samples were collected by N.E. and analyzed by
R.O.D. The instrumental error, uncertainties and calibration were conducted by R.O.D. with contributions from Z.A.K and
C.M. The automation and analysis software was developed by R.O.D. with contributions from K.P.B and M.C.C. The well
plate holder was designed by M.R. and R.O.D. and manufactured by M.R. The manuscript was written by R.O.D with
contributions from N.B.D, C.M. and Z.A.K. The project was supervised by Z.A.K.

## Acknowledgements

R.O.D. and Z.A.K. would like to acknowledge funding from SNF grant #200021_156581. Z. A. K. would like to acknowledge
Franz Conen and Emillano Stopelli for assistance with the initial set up of the droplet freezing assay. We are grateful to Dr.



James D. Atkinson for discussions and help with experiments during the preparatory phase of DRINCZ development. We
acknowledge technical assistance from Hannes Wylder. R.O.D. would like to thank Dr. William Ball for insightful statistical
discussions, Ellen Gute for performing camera tests and Michele Gregorini for assistance with the automation.
**Appendix A**
**Freezing bias by user**
The 20 SA water experiments were performed over a three month period by two users. The SA water was unaffected by aging
over this period as it originated from varying bottles distributed by the manufacturer (Sigma Aldrich). The user bias was
calculated the same way as the bias for all 20 experiments. The bias is relative to the median freezing temperature of the 4
corner wells obtained by the respective user. As can be seen in Fig. A2, the pattern of freezing bias is consistent regardless of
the user. This similarity indicates that the reported bias is instrumental and not user specific.

**Bias significance and correction**
To ensure that the observed bias is statistically significant, a two-sample, two-tailed $t$-test was performed. In particular, a
Welch's $t$-test was used due to the different number of samples between the combination of the 4 reference wells (20
experiments x 4 wells = 80 values) and each well (20 experiments x 1 well = 20 values) and the different variance of freezing
for each well (Derrick and White, 2016). In a Welch's $t$-test the location parameter of two independent data samples is assessed
as follows:
$$t = \frac{\overline{w}_{4ref} - \overline{w}_i}{\sqrt{\frac{s^2_{4ref}}{Nw_{4ref}} + \frac{s^2_i}{Nw_i}}}$$ (A1)
where $\overline{w}_{4ref}$ and $\overline{w}_i$ are the mean freezing temperature of the reference wells and an individual well, respectively. $s^2_{4ref}$ and
$s^2_i$ are the variances of freezing in the reference and the individual wells and $Nw_{4ref}$ and $Nw_i$ are the number of samples for
the reference wells and an individual well, respectively. The variance of the freezing temperature of SA water in each well is
shown as boxplots in the Appendix (Fig. A3). The temperature of approximately 30% of the wells was found to be statistically
different from the average freezing temperature of the 4 reference wells at the 95% confidence level, with a resultant mean
bias of 0.23 ˚C (Fig. 4b). Due to a fraction of wells with a statistically significant bias, a correction factor based on the mean
bias from the 20 SA water experiments is tested for all wells excluding the 4 corner wells used as the reference to avoid
overfitting the data. Of note, the reported bias is derived based on the freezing range of SA water from -16 to -26 °C. However,
based on the relatively constant spread in the temperature calibration data (see Fig. 3b), it is reasonable to assume that the bias
has a weak temperature dependence.



Although the freezing bias was shown to be representative when the SA water data was split in two (8 and 12 samples), it is
still necessary to validate its robustness on a larger sample size. In order to artificially increase the sample size of the
experiments, the bias was recalculated randomly such that only 90% or 18 of the experiments were used. The resultant bias
correction was then applied to the remaining 10% or 2 of the experiments and tested to see if the mean freezing temperature
of the bias corrected tray was closer to the reference freezing temperature of the 4 corner wells. This procedure was repeated
1000 times at random. The difference in the median freezing temperature (FF= 50 %) and 4 corner reference wells decreased
from 0.23 ˚C to 0.04 ˚C, while the standard deviation of the bias corrected data increased by 0.007 ˚C. Thus, the bias correction
performed as expected and reduced the bias in freezing temperature. Nonetheless, this improvement falls within the uncertainty
of the instrument, as discussed in Section 3.3 and is therefore not applied to DRINCZ measurements by default.

**Synoptic Summary Nov. 27th-30th**

The synoptic pattern over Europe on the 27th through 30th of November produced large variations in both temperature and air
mass origin at the SBO. As can be seen from the surface pressure maps shown in Fig. A5, an evolving cyclone tracked across
Northern Europe before occluding in the vicinity of Denmark. This cyclone produced strong warm advection at SBO on the
27th (see Fig. A7) in advance of the approaching cold front. As the cyclone began to fill over Southern Scandinavia, the cold
front stalled along the Alps and westerly flow continued at SBO from the 28th – 29th (Fig. A7). Farther west, the cold front
reached the Mediterranean where a secondary low developed along the remnant baroclinic zone (Fig. A6 panel c.). This
secondary low traversed Italy and rapidly intensified as it crossed the Adriatic Sea before entering the northern Balkans (Fig.
A6 panel d.). The secondary low and an amplifying ridge over the British Isles forced the cold front over SBO at 00Z on the
30th when cold air advection ensued over the SBO region (Fig. A7), as shown by the back trajectories (Fig. A8.e and f.).

**HYSPLIT back trajectories**

The Hybrid Single-Particle Lagrangian Integrated Trajectory model (HYSPLIT) (Stein et al., 2015) was run using the
interactive web portal (Rolph et al., 2017). The trajectories were calculated using 0.5˚ resolution and the trajectories were
initialized 1000, 2000 and 3000 meters above the model terrain height. Although the majority of snow mass growth has been
shown to occur between mountaintop and 1 km above the surface (Lowenthal et al., 2016), these heights were chosen due to
the coarse resolution of the model terrain height and the observed sensitivity of the back trajectories with height. HYSPLIT
was initialized using the 0.5˚ hourly Global Data Assimilation System (GDAS) archived database and the vertical velocity was
model based rather than isentropic.



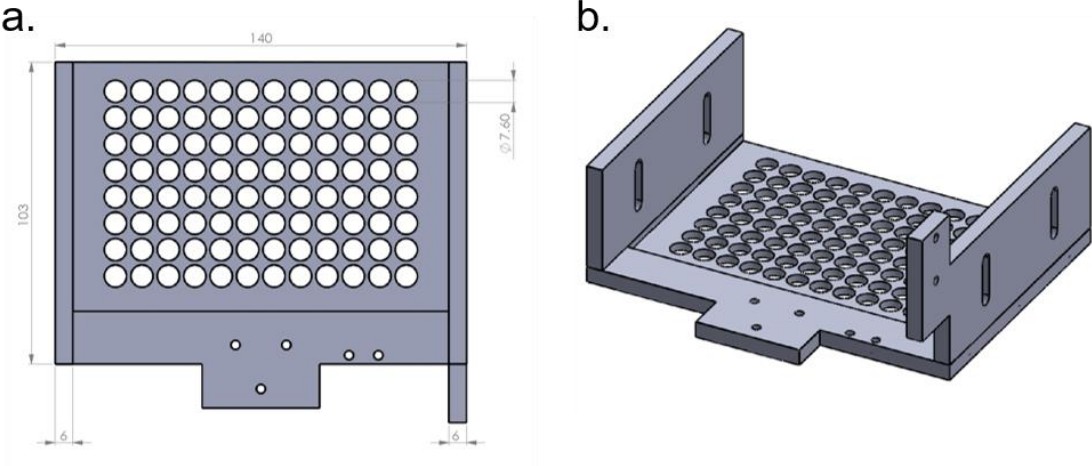


**Figure A1: Schematic of the 96-well tray holder from above (a) and the side (b), dimensions are in millimeters.**

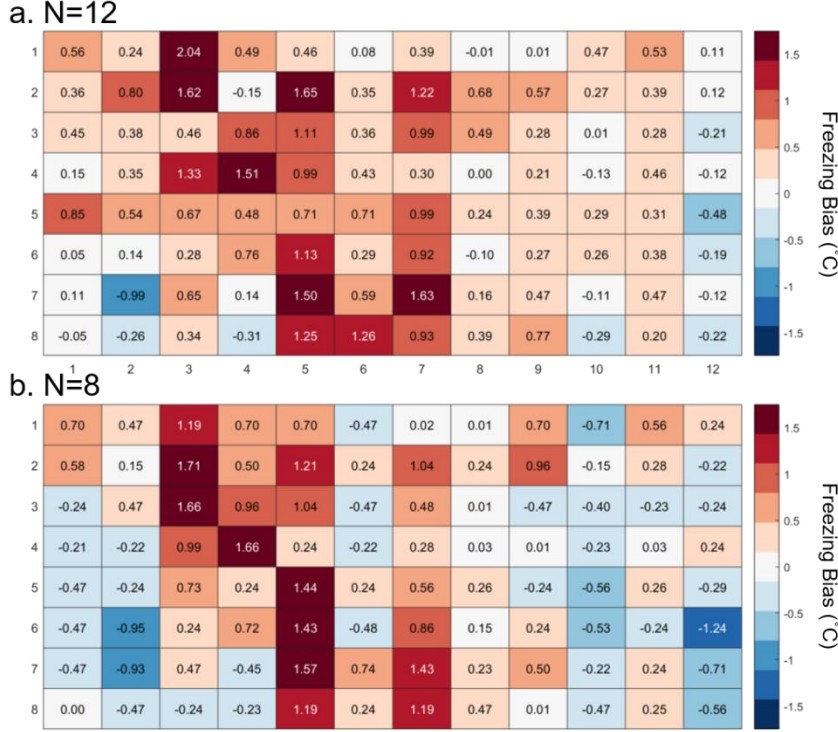


**Figure A2: (a) Bias in the freezing of SA water (˚C) based on the median value of each well over 12 experiments and (b) 8 experiments**
**relative to the median temperature of freezing for the 4 corner wells used during the temperature calibration. A positive (negative)**
**bias indicates that the wells experience a warmer (colder) temperature than the four corner wells used for temperature calibration**
**and therefore freeze at lower (higher) temperatures than reported.**




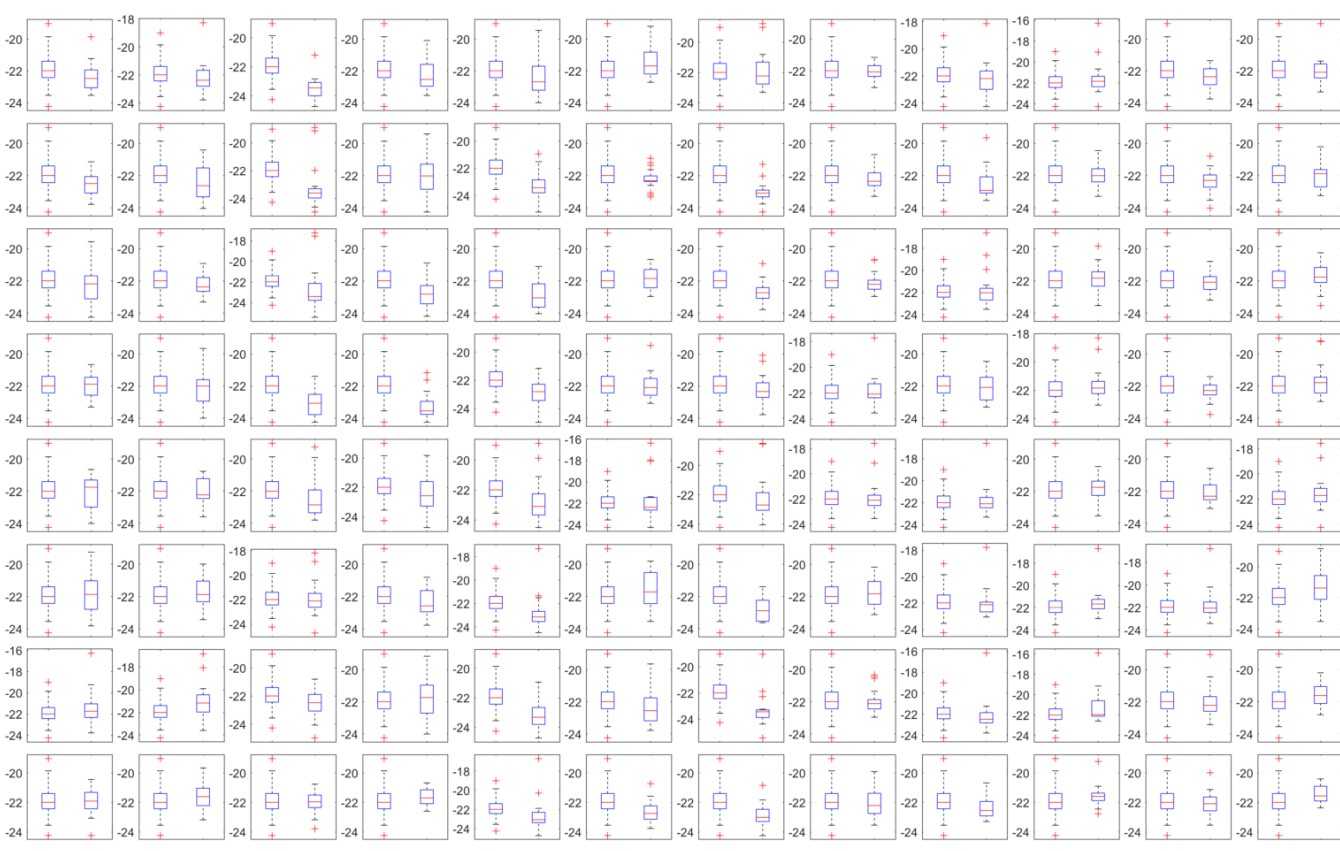


**Figure A3. A side by side comparison of box plots for the freezing temperatures of the 20 SA water experiments of the reference wells (left box) and the well represented by the location (right box) of each subplot. The median (red line), inter-quartile range (blue box), extreme values not considered outliers (whiskers) and outliers (red crosses) are shown as a function of temperature in ˚C (y-axes).**





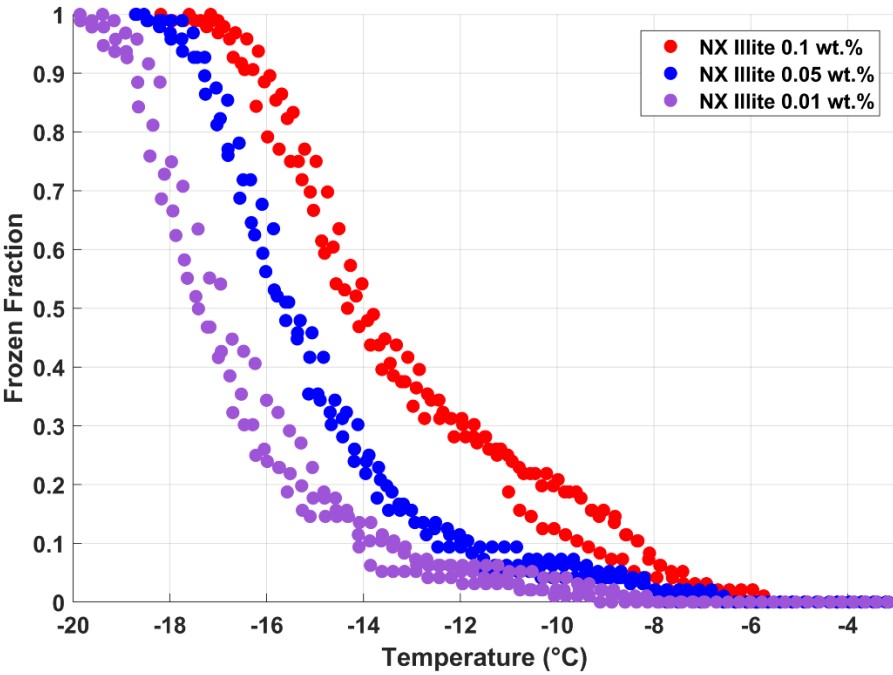


**Figure A4. Frozen fraction curves of solutions of 0.01 wt. % (magenta dots), 0.05 wt. % (red dots) and 0.01 wt. % (purple dots) of**
**NX-illite run in triplicates.**

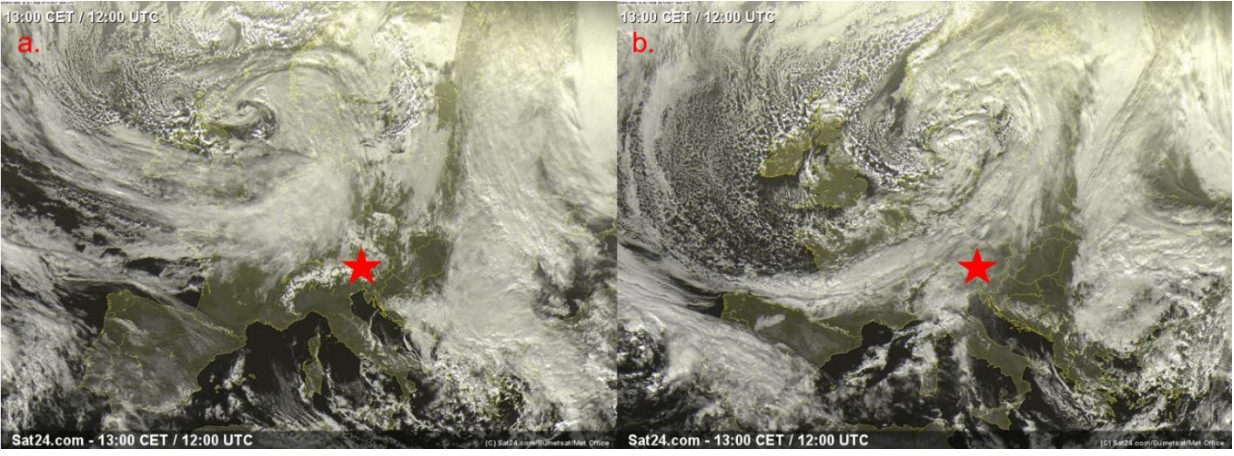


**Figure A5: Visible satellite image of the storm system impacting the SBO (red star) taken at 1200UTC on (a) Nov. 27th and (b) 28th.**
**Images courtesy of Sat24.com/Eumetsat/Met Office (http://www.sat24.com/history.aspx).**



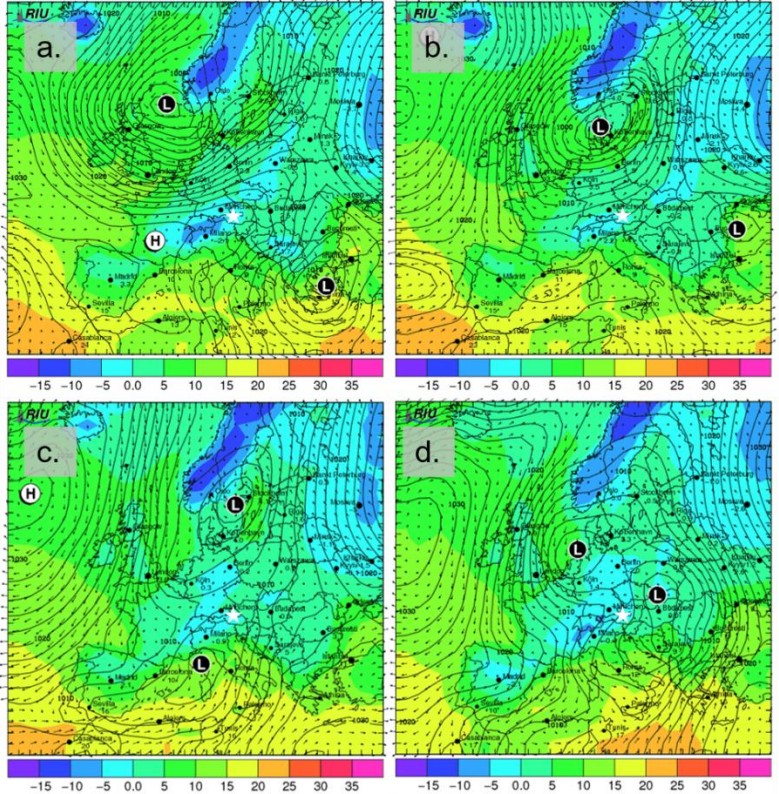

**Figure A6:** Forecasted surface pressure in hPa (black contours), 2 meter surface temperature in ˚C (color fill), and wind vectors in m/s (black arrows) for 12 UTC on (a) 27ᵗʰ, (b) 28ᵗʰ, (c) 29ᵗʰ and (d) 30ᵗʰ. Forecasts are based on model runs initialized on 00 UTC of the day of interest (12 hours before shown values). Surface low and high pressure centers are indicated with L and H, respectively. The location of SBO is shown by the white star. Images are taken and adapted from the Rhenish Institute for Environmental Research at the University of Cologne (http://www.uni-koeln.de/math-nat-fak/geomet/eurad/index_e.html).



590

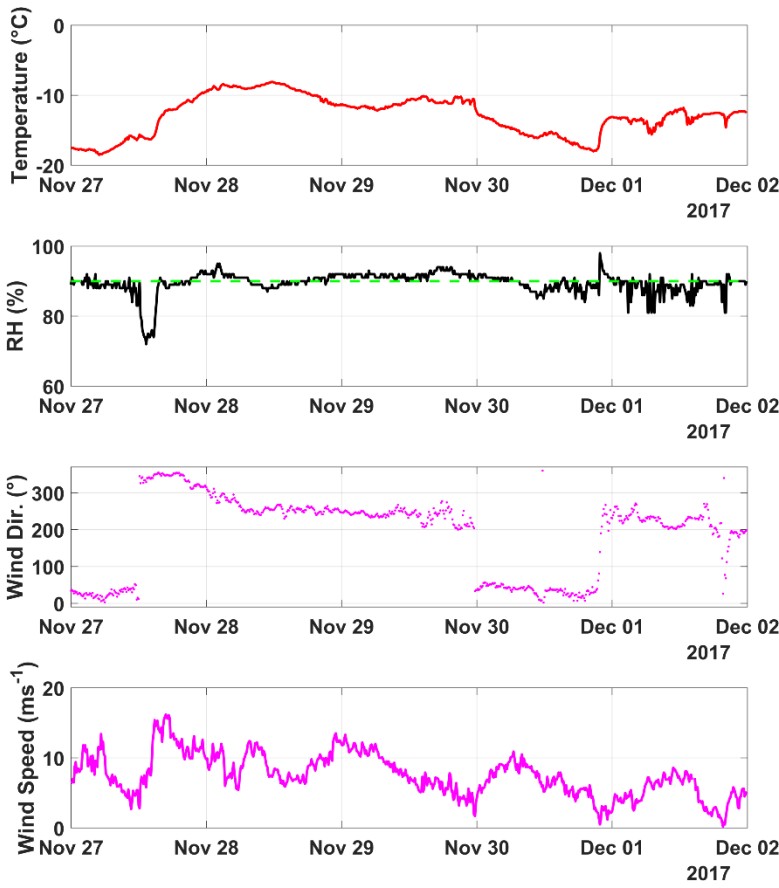

591

**Figure A7: (top panel) Temperature (˚C), (top-middle panel) humidity (%), (bottom-middle panel) wind direction (˚) and (bottom panel) wind speed (ms⁻¹) as a function of date spanning from the 27ᵗʰ of November to the 2ⁿᵈ of December (in UTC). The humidity when cloud is present at SBO (90%) is shown (dashed green line).**

**Figure A8:** (a) 84-hour HYSPLIT back trajectories from the Sonnblick Observatory initialized on 00 UTC on the 27th, (b) 12 UTC on the 28th, (c) 06 UTC and (d) 18 UTC on the 29th, and (e) 00 UTC and (f) 12 UTC on the 30th of November. The blue, green and red lines represent 8 ensemble back trajectories initialized 1000 m, 2000 m and 3000 m above the model terrain height, respectively. The two lower panels in each subplot show the back trajectory height in units of pressure (hPa) and rainfall (mm) as a function of time (in 6 hourly intervals) as a function of pressure (in hPa).





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
