# Peer review of "Development of the DRoplet Ice Nuclei Counter Zürich (DRINCZ)"

_Atmospheric Measurement Techniques, 2019_

## Referee Comment (RC1) · Gabor Vali (Referee) · 30 Jul 2019

Reviewer comments on **"Development of the DRoplet Ice Nuclei Counter Zürich (DRINCZ): Validation and application to field collected snow samples"** by  Robert O. David, Maria Cascajo Castresana, Killian P. Brennan, Michael Rösch, Nora Els, Julia Werz, Vera Weichlinger, Lin S. Boynton, Sophie Bogler, Nadine Borduas-Dedekind, Claudia Marcolli, Zamin A. Kanji

Gabor Vali

**General:**

This paper is a good addition to the literature on INP measurements. Not fundamentally new, but every implementation of the drop-freezing technique, or of any other method, brings new challenges and new ways of solving them. The authors deal with those challenges reasonably well. This paper stands out with its focus on evaluating instrument-caused uncertainties. There are some parts of that evaluation that would benefit from a second look.

**Detailed comments:**

line numbers

| | |
|---|---|
| 41-57 | In describing different approaches to INP detection it is useful to separate those that examine air samples and those that take water samples. |
| 81 | Does Bigg (1953) contain data and hail and snow samples? Please check. |
| 88 - 89 | The goals are stated in overly broad terms. The proposed measurement are expected to be relevant to MPC clouds but no claim should be made that they examine those responsible for ice formation in those cloud. There are other elements to that story beyond the INP measurements. Also, to what degree can these measurements illuminate 'fundamental understanding' of ice nucleation? |
| 97 | In addition to those cited, a design much like the one in this paper was described by Vali (1995; Principles of ice nucleation. Chapter 1 in: "Biological Ice Nucleation and Its Applications", R. E. Lee Jr, G. J. Warren, and L. V. Gusta, Eds., APS Press, The American Phytopathological Society., St. Paul, Minnesota, USA. 370 pp.; ISBN: 0-89054-172-8). |
| 103 | Is the foil seal enough to exclude ethanol vapors from getting into the samples and thereby producing a freezing point depression? |
| 123 -> | Since the position of the sample tray is fixed, and so is its dimension, why is such an elaborate process necessary for identifying the well locations? Also, if done this way, to what extent does perspective from the camera lens distorts the circular shape of the wells near the edges of the tray? |
| 131-132 | The meaning of " .... centered at the edge ... as the well center." could probably be |

clarified better.

| | |
|---|---|
| 135 | Random order and sorting seem unnecessary with the fixed geometrical arrangement of this setup. What is the rationale here? |
| 152 | Isn't the first instant of intensity change over a threshold magnitude sufficient to detect nucleation? If not so, why not? What possible reason exists for a significant peak in the signal, comparable to that caused by nucleation, prior to nucleation? |
| 157 | What is meant by "all recorded images"? |
| Fig 2. | It is unclear to me what mean intensity and normalized intensity refer to. Is it an average within the circle for a given well? Are they for a given well over repeated trials? Are the averages over many wells? |
| 190 | What is 'maximum standard deviation'? |
| 196 -> | The work here described in Section 3.2 is certainly well directed and quite extensive. However, it is surprising that nucleation temperatures of SA water are used instead of direct temperature measurements. It is the temperature of the water before, and at the instant, of nucleation that is most relevant. Direct temperature measurement of the water in the wells is not without its own difficulties (locating the sensors in the wells, sensor lead errors, etc.) but the variations in nucleation temperatures from well to well, even for SA water or other similar sample, are bound to be adding uncertainty to the calibration. What governed the decision to use nucleation temperatures to evaluate bias across the well-plate? To assess the quality of this approach it would be useful to know how much variation in nucleation temperatures was observed for any given well within the 20 repeat tests. The two sources of variations - within a given well and among different wells - should be both presented and the sufficiency of the use of the median for each well thus evaluated. |
| 226 | Section numbering is off. |
| 244 | This standard deviation refers to the distribution of observed freezing temperatures among wells? Again, please distinguish between single well repetitions and variations among wells. |
| 246 | The 50% fraction corresponds to the steepest point on the FF curve for SA water. But this is not a general result; other samples may have no such correspondence between the two measures. |
| 258 - 259 | Fig. 4a shows, as expected, that the standard deviation varies according to the slope of the FF curve (sample size effect). Assigning this patters to the influence of ethanol circulation is likely to be incorrect. More general point: to what extent in ethanol circulation predictable? This is a valid question in light of the flow being turbulent with the level control adding pulses of liquid. |

304 -> The background correction via Eq. (10) is valid, but it is surprising that the correction is finally presented in terms of FF, via Eq. (11). Fitting a correction equation to $k_{bg}(T)$ would be more direct and more readily applicable to a variety of samples with different volumes and/or dilutions.

342 -> Section 4.2 is well done. It is a good demonstration of the DRINCZ's capabilities. Was background correction applied?

373 -> Sections 4.3 to 4.5 introduce a topic beyond the description of the instrument. As has been amply shown in the extensive literature on the topic, analyses of snow samples are valid tools as inputs to the analyses of cloud processes, but with the attendant complicating factors partially discussed here. That current results vary within the range reported for other such measurements is due does demonstrate that the sampling techniques were adequate and that the atmosphere is relatively conservative in the range of INP contents of snow. They do not substantially reinforce the validation of the instrument per se; that validation is more clearly supported by the calibrations and by the illite sample results. It is not stated (or it escaped me) whether the freezing analyses were done in the field or in the laboratory. This would be relevant to possibly show that the instrument is rugged enough for field use and that different setups do (or do not) effect the results.

---

## Referee Comment (RC2) · Anonymous Referee #2 · 31 Jul 2019

The comment was uploaded in the form of a supplement:
https://www.atmos-meas-tech-discuss.net/amt-2019-213/amt-2019-213-RC2-supplement.pdf

---

## Author Comment (AC1) · 10 Oct 2019

Reviewer comments on "**Development of the DRoplet Ice Nuclei Counter Zürich** (**DRINCZ**):Validation and application to field collected snow samples" by Robert O. David, Maria Cascajo Castresana, Killian P. Brennan, Michael Rösch, Nora Els, Julia Werz, Vera Weichlinger, Lin S. Boynton, Sophie Bogler, Nadine Borduas-Dedekind, Claudia Marcolli, Zamin A. Kanji By Gabor Vali

**General:**

This paper is a good addition to the literature on INP measurements. Not fundamentally new, but every implementation of the drop-freezing technique, or of any other method, brings new challenges and new ways of solving them. The authors deal with those challenges reasonably well. This paper stands out with its focus on evaluating instrument-caused uncertainties. There are some parts of that evaluation that would benefit from a second look.

We would like to thank Gabor Vali for his positive and very helpful comments and respond to the individual concerns below. Reviewer comments have been reproduced in bold typeface, and author responses are in regular font. All line numbers in authors' response refer to the revised manuscript.

**Detailed comments:**

**41-57 In describing different approaches to INP detection it is useful to separate those that examine air samples and those that take water samples.**

In lines 41-56, we are describing single particle methods, which use aerosolized samples that can either be dry dispersed or atomized from a suspension. We now clarify this in lines 51-52. Thus, they can be used to investigate air samples, suspensions or seawater samples. Furthermore, we add "*immerse the samples in water and*" to line 60 to show that bulk methods use samples suspended in solution.

**81 Does Bigg (1953) contain data and hail and snow samples? Please check.**

Thank you for pointing this out. Indeed, Bigg, (1953) does not use or describe a bulk method to investigate INPs. The citation has been removed from line 83 (revised manuscript).

**88 - 89 The goals are stated in overly broad terms. The proposed measurement are expected to be relevant to MPC clouds but no claim should be made that they examine those responsible for ice formation in those cloud. There are other elements to that story beyond the INP measurements. Also, to what degree can these measurements illuminate 'fundamental understanding' of ice nucleation?**

We have reformulated the sentence to more explicitly describe the motivation behind the development of DRINCZ. The sentence now reads (90-94): "In order to further quantify the variability of ambient INP concentration relevant for ice formation in MPCs and increase the understanding of the ice nucleation ability of laboratory and field collected samples, we developed and characterized the DRoplet Ice Nuclei Counter Zurich (DRINCZ). DRINCZ is a drop freezing instrument to investigate ice nucleation at temperature conditions between - 25 °C and 0 °C, representative for MPCs."

97 In addition to those cited, a design much like the one in this paper was described by Vali (1995; Principles of ice nucleation. Chapter 1 in: "Biological Ice Nucleation and Its Applications", R. E. Lee Jr, G. J. Warren, and L. V. Gusta, Eds., APS Press, The American Phytopathological Society., St. Paul, Minnesota, USA. 370 pp.; ISBN: 0-89054-172-8).

Thank you, we have now added the citation to the text (line 101).

**103 Is the foil seal enough to exclude ethanol vapors from getting into the samples and thereby producing a freezing point depression?**

This is a valid concern but we believe is not an issue in our experiments. The comparison with literature values of NX-Illite as shown in Fig. 9 do not show a significant bias when compared to techniques that use a sealed cooling block where exposure to ethanol vapors is not an issue. Furthermore, the foil seals used with DRINCZ are non-permeable, suggesting that the probability of ethanol vapors entering the wells would be negligible. To clarify this, we have added "non-permeable" to the description of the foil (line 107).

**123 -> Since the position of the sample tray is fixed, and so is its dimension, why is such an elaborate process necessary for identifying the well locations?**

As DRINCZ is meant to be a field deployable instrument, the camera mounting location is variable. Even if it is mounted as reproducibly as possible, there are still some variations that can lead to issues when using a fixed well location. Therefore, the code looks for the wells instead of hard coding their locations. This justification is provided at the start of section 2.1.1, lines 129-132 (124-127 in original manuscript).

**Also, if done this way, to what extent does perspective from the camera lens distorts the circular shape of the wells near the edges of the tray?**

This is a valid point and some edge effects can be seen in Fig. 2a. Nevertheless, the change in light transmission through the well is still significant enough to be detected by the camera to overcome the distortion and shading from edge effects and therefore provide an accurate freezing temperature.

**131-132 The meaning of " .... centered at the edge ... as the well center." could probably be clarified better.**

Thank you for pointing out that this is confusing. We have therefore reworded the sentences and the preceding sentence to state:" *The CHT first identifies pixels along regions of large gradients in brightness to identify pixels at the edge of the well. To determine the center of each well, the algorithm draws circles of varying diameter (ranging between 15 and 30 pixels in diameter, which corresponds to the observed diameters of a well in terms of pixel number) around these edge pixels and classifies the pixel intersecting the largest number of circles as the well center." (lines 135-138)*

**135 Random order and sorting seem unnecessary with the fixed geometrical arrangement of this setup. What is the rationale here?**

As described on lines 129-132 (lines 123-125 in original manuscript), the location of the camera is not fixed in the current setup and can be removed for easier packing and shipping. Therefore, the well locations are not hard coded into the software but rather identified using the CHT. However, future versions of DRINCZ could attempt to have a perfectly reproducible camera location and mounting system so that this would not be needed anymore.

**152 Isn't the first instant of intensity change over a threshold magnitude sufficient to detect nucleation? If not so, why not? What possible reason exists for a significant peak in the signal, comparable to that caused by nucleation, prior to nucleation?**

The threshold proved to be necessary to account for fluctuations in the light transmission through a well due to turbulence and air entering the ethanol bath. The noise arising due to these fluctuations can be seen in panels b and c of Fig. 2.

**157 What is meant by "all recorded images"?**

We simply mean all images, and have removed "recorded" from the revised manuscript so as to simply mean all of the images of a well (see line 165)

**Fig 2. It is unclear to me what mean intensity and normalized intensity refer to. Is it an average within the circle for a given well? Are they for a given well over repeated trials? Are the averages over many wells?**

The mean intensity  $(I_t)$  is just the average value of all of the pixels in a single well and is explained in lines 157-158. Therefore, there is a mean intensity for each well at every temperature (every image). Similarly, there is a normalized value  $Z'_t$  for each well at every temperature (each image, see lines 163-168).

To clarify this, we have now added "of *a single well*" to the caption of Figure 2b and "*for the same well as in b*" to the explanation of the caption for Figure 2c.

**190 What is 'maximum standard deviation'?**

We have now removed maximum standard deviation from the sentence and added a new sentence to explain the maximum standard deviation which reads: "The maximum standard deviation taken as the temperature difference between the temperature fit and the individual well temperature was  $\pm 0.6$  °C." on lines 203-205.

196 -> The work here described in Section 3.2 is certainly well directed and quite extensive. However, it is surprising that nucleation temperatures of SA water are used instead of direct temperature measurements. It is the temperature of the water before, and at the instant, of nucleation that is most relevant. Direct temperature measurement of the water in the wells is not without its own difficulties (locating the sensors in the wells, sensor lead errors, etc.) but the variations in nucleation temperatures from well to well, even for SA water or other similar sample, are bound to be adding uncertainty to the calibration. What governed the decision to use nucleation temperatures to evaluate bias across the well-plate?

To directly measure the temperature of each well during an experiment, 96 thermocouples would be needed, which all can vary in temperature by about  $\pm 2^{\circ}$ C if they are not calibrated accurately. Alternatively, if the same thermocouple were used for all wells, 96 freezing runs would be needed to obtain only one temperature measurement for each well. Such a procedure is not feasible. Therefore, freezing runs performed with SA water were averaged, such that random variability cancelled out and systematic bias added up.

To assess the quality of this approach it would be useful to know how much variation in nucleation temperatures was observed for any given well within the 20 repeat tests. The two sources of variations - within a given well and among different wells - should be both presented and the sufficiency of the use of the median for each well thus evaluated.

We completely agree. The spread in the freezing temperature of the SA water in individual wells was indeed included in the original manuscript in the Appendix as Fig. A3, but was not mentioned in the main text. Although the distribution of freezing in the wells varied, the median was chosen as the most representative due to its definition as the center value of the distribution. Thus, it should be less sensitive to outliers than the mean. We have now added a reference to this figure on line 218 and reordered the Appendix figures accordingly, as such it is now Fig. A2.

**226 Section numbering is off.**

Thank you for pointing this out. We have now renumbered the section to be consistent with the rest of the manuscript (see line 242).

**244 This standard deviation refers to the distribution of observed freezing temperatures among wells? Again, please distinguish between single well repetitions and variations among wells.**

We have now added (lines 259-261) that the standard deviation here refers to "the standard deviation in the observed freezing temperatures of the SA water experiments across all wells"

**246 The 50% fraction corresponds to the steepest point on the FF curve for SA water. But this is not a general result; other samples may have no such correspondence between the two measures.**

This is absolutely true, but we chose the 50 % FF here as this is the most probable temperature at which the SA water freezes and therefore represents the best estimate of the bulk freezing properties of the water with a reduced influence from outliers and contamination. To clarify the reviewer concern we have now added the sentence (lines 263-264): *"Furthermore, by using the 50 % FF the influence of contamination and outliers is reduced."*

**258 - 259 Fig. 4a shows, as expected, that the standard deviation varies according to the slope of the FF curve (sample size effect). Assigning this patters to the influence of ethanol circulation is likely to be incorrect.**

Figure 4 shows that the deviation (bias) of each well in freezing temperatures from the median (panel a) and mean value (panel b) exhibit a non-random pattern. We see the ethanol circulation as the most likely explanation for such a pattern as the cooled ethanol circulates around the tray in a clockwise direction as indicated by the arrows in Fig. 4. We now clarify this on lines 222-224 of the revised manuscript.

**More general point: to what extent in ethanol circulation predictable? This is a valid question in light of the flow being turbulent with the level control adding pulses of liquid.**

In regards to the impact of the ethanol pulses on the ethanol circulation, it is important to point out that the pulses add very small volumes of ethanol and therefore likely have little impact on the circulation. In contrast, the change in bath level due to contraction during cooling likely has larger impacts on the flow in the bath due to changes of the exposed internal surface area of the bath. Therefore, the addition of ethanol likely makes the bath circulation more consistent. Regardless, no impact on the circulation was observed when observing the bath by eye with or without the use of the bath leveler.

**304 -> The background correction via Eq. (10) is valid, but it is surprising that the correction is finally presented in terms of FF, via Eq. (11). Fitting a correction equation to $k_{bg}(T)$ would be more direct and more readily applicable to a variety of samples with different volumes and/or dilutions.**

Indeed we use the method described by the reviewer, i.e.  $k_{bg}(T)$  is used when correcting for the freezing background. The conversion to *FF* is just used to demonstrate how the background influences the *FF*.

**342 -> Section 4.2 is well done. It is a good demonstration of the DRINCZ's capabilities. Was background correction applied?**

Thank you for pointing this out. The results presented in section 4.2 are background corrected so we have added: *"and background corrected (using Eq. 11)"* to the sentence (lines 373-374).

373 -> Sections 4.3 to 4.5 introduce a topic beyond the description of the instrument. As has been amply shown in the extensive literature on the topic, analyses of snow samples are valid tools as inputs to the analyses of cloud processes, but with the attendant complicating factors partially discussed here. That current results vary within the range reported for other such measurements is due does demonstrate that the sampling techniques were adequate and that the atmosphere is relatively conservative in the range of INP contents of snow. They do not substantially reinforce the validation of the instrument per se; that validation is more clearly supported by the calibrations and by the illite sample results. It is not stated (or it escaped me) whether the freezing analyses were done in the field or in the laboratory. This would be relevant to possibly show that the instrument is rugged enough for field use and that different setups do (or do not) effect the results.

We acknowledge that the measurements of field collected samples do not act as a validation of the technique. Rather they are added to the manuscript to show that the technique can be applied to field collected samples while providing the scientific community with additional observations of INP concentrations collected in snow samples.

In this case DRINCZ was not deployed in the field but the samples were shipped frozen to the laboratory in Zurich where they were stored frozen until the experiments were conducted. We have now adapted lines 403-404 to clarify that the samples were shipped frozen and the measurements with DRINCZ were conducted in Zurich by changing the sentence to read as: *"The samples were shipped and stored frozen until processed with DRINCZ at the Atmospheric Physics laboratory at ETH Zurich, to minimize any bacterial growth or changes due to liquid storage* (Stopelli et al., 2014)."

**References:**

Bigg, E. K.: The formation of atmospheric ice crystals by the freezing of droplets, Q. J. R. Meteorol. Soc., 79(342), 510–519, doi:10.1002/qj.49707934207, 1953.

Stopelli, E., Conen, F., Zimmermann, L., Alewell, C. and Morris, C. E.: Freezing nucleation apparatus puts new slant on study of biological ice nucleators in precipitation, Atmospheric Meas. Tech., 7(1), 129–134, doi:10.5194/amt-7-129-2014, 2014.

---

## Author Comment (AC2) · 10 Oct 2019

**Review of "Development of the Droplet Ice Nuclei Counter Zürich (DRINCZ): Validation and application to field collected snow samples" by David et al.**

**General comment**

**In this manuscript the authors describe and characterise a large volume immersion mode drop assay (DRINCZ). The authors thoroughly characterise the horizontal temperature gradient across the 96 well plate in the system and recommend a correction which is of use to other instrumental setups. The authors report a ± 0.9 ˚C uncertainty for DRINCZ and go on to validate the instrument by comparing to literature data of NX-illite. A field study investigating snow melt samples is also undertaken which shows agreement (mostly) with previous snow melt measurements. The authors then relate the INP concentrations measured to airmass trajectories and propose scavenging of INP by precipitation led to the lowest INP concentrations measured.**

**This manuscript is well written and presents results which are of interest to the ice nucleation community. The manuscript is in the scope of AMT and I support its publication after the following comments have been properly addressed.**

We thank the reviewer for the positive recommendation and for raising several points that we now address individually below and in the revised manuscript to make the paper clearer. Reviewer comments reproduced in bold and author responses in regular typeface. All line numbers in authors' response refer to revised manuscript.

**Major comment**

**Although I like the manuscript and find the results of use to the ice nucleation community, I am unclear on the novelty of this instrument compared to others that have already been presented. The authors have acknowledged that the technique is based on the design of previous instruments. Is the method used to characterise and correct the horizontal gradients in the plate the only novelty? If so, I suggest this is made clearer in the final manuscript or that the unique traits of this instrument are further clarified.**

We thank the reviewer for pointing this out. Indeed, the instrument is quite similar to previously developed drop freezing assays. New aspects are a method for determining horizontal gradients across the well plate and a fully automated data analysis, which only requires the user to enter a folder path name into a MATLAB function. We have now added the ease of data analysis as one of the benefits of DRINCZ by stating in the abstract (line 23): *"with a user friendly and fully automated analysis procedure."* Unfortunately, it is difficult to directly compare the ease of use and data analysis developed for DRINCZ relative to similar setups based on published papers, so we cannot be more specific.

**The horizontal gradients of the plate have been characterised but the vertical gradients within the wells have not been explored. These should be discussed in the text. A reference to Beall et al. (2017) would be appropriate as they characterise the gradient within 50 µL droplets within wells with a similar profile (PCR plate).**

It is very difficult to measure the vertical bias in a 50 µL well and this is also acknowledged in Beall et al. (2017). It is important also to note that in the setup of Beall et al. (2017), the polypropylene well tray is in contact with an aluminum block rather than the ethanol itself. This can lead to gradients due to the block, which would be negligible in our setup where the tray is in direct contact with the ethanol. Nevertheless, we have mentioned that the bath leveler can help reduce this issue on lines 176-180. We now reference Beall et al. (2017) in the revised manuscript to point out the possibility of vertical gradients in the wells and write

that we attempt to avoid them by ensuring that the entire volume of solution inside the well is surrounded by ethanol. The addition to the text reads: *"It has been shown that large vertical gradients of up to 1.8 °C can exist between the bottom of a well and the air above it in block-based drop freezing setups (Beall et al., 2017). We anticipate vertical gradients to be reduced in DRINCZ due to the direct contact between the cooling medium (ethanol) and the well tray when the ethanol levels remains constant during cooling. Therefore, we incorporated a bath leveler composed of a level sensor and solenoid valve to ensure that the ethanol level remains constant."* (lines 176-180).

**Figure 9 displays data for NX-illite dilutions. I find the text a little misleading in presenting the data as though there are only a "few" outliers for the 0.01wt% dilution. The vast majority of data for all three triplicates for this dilution give higher freezing temperatures than higher weight percent suspensions (at the same value of ns). This is in contradiction to what we expect of ns. I believe the source of this error is different to the uncertainties characterised in previous sections as the data is consistently offset to higher freezing temperatures. This issue is not seen (in most cases) in the dilutions for the snow melt study and suggests this discrepancy may be material dependent and related to the distribution of particles. Although the authors do mention this issue, the extent of the discrepancy between dilutions is glossed over in the text. I must stress that I do not believe this inconsistency in the dilutions is a result of an error in the instrument but rather an error as a result of the material or sampling method. With this said, the results should be presented in the text to acknowledge the true extent of this discrepancy.**

The observed difference in the $n_{sBET}$ values at the lowest weight percent compared with the higher concentrations falls within the uncertainty of the instrument (± 0.9 °C). Moreover, there is considerable variability between the triplicates performed with the same suspension concentrations. This can be seen in Fig. A4, where the *FF* curves of all NX-illite DRINCZ experiments are shown. In addition, we have updated Fig. 9 so that the difference between the triplicates of the NX-illite suspensions can be seen more easily. Nevertheless, we acknowledge that $n_{sBET}$ of the 0.01 wt% NX-illite suspension is constantly above the $n_{sBET}$ of the 0.05 and 0.1 wt% NX-illite suspensions. One reason for this might be that very few random freezing events occurring at warm temperature in the higher diluted sample may constantly increase cumulative active site densities to lower temperatures. However, based on the available data, we cannot exclude an effect due to dilution of a single stock suspension, which can lead to a bias compared with preparing suspensions of each concentration separately. We now discuss both possibilities in section 4.2 and reference the Harrison et al. (2018) study where issues arising from diluting a single stock suspension are discussed in detail. Nonetheless, we need to emphasize, that more investigations would be needed to establish the significance of the increased $n_{sBET}$ observed at the lowest suspension concentration given the random variability between repetitions in such freezing experiments.

**Minor comments**

**Line 67-77: Dilution is not the only means of changing the measurable range of INP. Concentrating the particles per droplet can also extend the range. I suggest this is added to the discussion.**

This is a valid point. We add on line 71-73 after "aliquot": *"Alternatively, to explore freezing towards warmer temperatures, field samples (e.g. rain or snow samples) can be concentrated by evaporating a part of the sample water."*

**Line 116: At what temperature does the ethanol bath start at and what temperature does it end, i.e. 0 ˚C to -30 ˚C. In addition to this, if the sample is added to an ethanol bath at 0 ˚C (as suggested by line 165) is the sample allowed time to equilibrate? If not, this could lead to thermal gradients not just horizontally across the plate but vertically in the wells (see major comment). I suggest adding information on the cooling profile of the bath to this section.**

The wells are left to equilibrate to the bath temperature at 0°C for one minute before the experiment and cooling ramp is initiated. We have now added to the text on lines 107-108: *"The well tray is placed in the tray holder (Fig. A1) and left to rest for 1 min at 0°C before the experiment is started."*

**Section 2.1.2: This describes the detection of freezing events in wells. Is this similar to other methods, e.g. (Stopelli et al., 2014)? Clarify what is different.**

There is no appreciable difference in the detection method for identifying a freezing event relative to Stopelli et al. (2014). We have now added "*Similar to Stopelli et al, (2014)*" on line 123. However, rather than using a fixed intensity change as done by Beall et al, (2017), we use a normalized threshold of 0.6 as explained in the section. We have now clarified this difference on lines 168-170 by stating: "*…rather than relying on a fixed change in light transmission through the well as done by other drop freezing setups (Beall et al., 2017). This ensures that the initial freezing detection is independent of the absolute change in light transmission through a well."*

**Section 2.2: What is the error of the sensor? What will be the fluctuation in the ethanol level? Do the authors consider it negligible?**

There is no appreciable error in the sensor as it is a binary switch that either detects contact with or without the ethanol. As the sensor triggers the opening and closing of the solenoid valve to allow ethanol to flow into the bath, we expect fluctuations in the ethanol bath to be negligible.

**Line 182-183: K type thermocouples can have large uncertainties, commonly ± 2.2 ˚C, compared to other thermocouple types (e.g. T type). Were these K type thermocouples calibrated other than by the manufacturer? What is the error of these? I suggest showing these errors in the figures (or an example of the errors).**

Indeed, there can be differences between different thermocouples. That is why we did the temperature calibration using the same thermocouple in all five test locations. Therefore, the observed difference in the well temperature is based on the same thermocouple and not sensor dependent. As the temperature error reported is based on one thermocouple, the differences that we report can be attributed to the locations of the wells in the tray. We did not calibrate the thermocouple in-house but rather compare the thermocouple temperature to the bath temperature of the chiller which is measured by a PT-100 temperature sensor. As such we have clarified this in the text by adding: "*The same thermocouple was used for all the well temperature measurements to avoid biases between different thermocouples.*" to lines 196-197

**Line 183-184: Were the wells completely filled with ethanol or 50 µL? If completely filled does this represent the gradients that would be present in the wells and plate in a typical 50 µL experiment?**

The wells were filled with 50 µL of ethanol to reproduce the experimental procedure used for DRINCZ freezing runs. We have now added *"50 µL of ethanol …"* to the text on line 197.

**Section 3.2: The characterisation of the horizontal gradient across the plate is very useful for the community. However, has the vertical gradient within the wells been considered (see major comment)? This system uses a similar well profile to that used by Beall et al. (2017) who found that a vertical stratification of 0.5 ˚C can be found in wells in which the headspace (air above the wells) is ≥6 ˚C warmer. Please discuss this in the text and reference Beall et al. (2017) where appropriate.**

It is very difficult to measure the vertical bias in a 50 µL well and this is also acknowledged in Beall et al. (2017). It is important to note that in the setup of Beall et al. (2017), the polypropylene well tray is in contact with an aluminum block rather than the ethanol itself. This can lead to gradients due to the block, which would be negligible in our setup where the tray is in direct contact with the ethanol. Nevertheless, as we now state in the response to the major comments, the bath leveler helps to reduce this issue (see response in major comments and revised manuscript line 175-180).

**Also, you use the median freezing temperature of SA water to determine the offset in temperatures across the plate. Is there not a random probability of the SA water freezing at different temperatures without a temperature bias to start with? Would there also not be uncertainties due to accidental contamination in the individual wells during the setup of the experiment? Does this not create uncertainty in this experiment? What was the rationale for using SA water? If you were to use freezing temperatures (rather than direct measurements of the wells) then would something that froze more consistently at the same temperatures be a better standard to use, i.e. pollen has a narrow window for freezing.**

We completely agree that there is a stochastic component to the freezing of the SA water. However, by averaging several experiments, the random variations cancel out while systematic bias adds up. Moreover, we calculate the precision of the instrument using the standard deviation of the temperature required for 50 % of the wells to freeze (see Section 3.3), which is less affected by random variability. In addition to minimize the effect of stochasticity, the 50% FF should also reduce the influence of contamination. Therefore we have added a sentence to Section 3.3 on line 263-264 stating: *"Furthermore, by using the 50 % FF the influence of contamination and outliers is minimized."* We chose SA water as the bias and precision standard for two reasons. First, the SA water is used in the majority of the experiments to prepare or dilute the samples. Therefore, its freezing curve needs to be well known for background correction. Moreover, we use SA water as a reference sample to control the performance of DRINCZ and the constancy of the background. Therefore, we accumulated a large number of DRINCZ experiments with SA water available for closer analysis. Second, SA water has the lowest accessible freezing temperatures and at the lowest temperatures we expect the largest bias since the gradient between the air and the bath temperature is maximized.

**Section 3.3: It is unclear to me why you assess the uncertainty of the instrument and combine this with the variability in the freezing temperature of SA water at this point. The water baseline in other studies, e.g. field-based studies, could potentially be worse (or better) than what you have done in these experiments. Should the experimental error as a result of the water impurities not be considered separately in respect to the particular experiment/environment?**

Indeed, we use the freezing experiments performed with SA water for two purposes. First, in the laboratory we need to know the background freezing due to impurities present in the SA water for the samples that we prepare, collect or dilute with SA water (see Section 4.1). In the field, we also conduct background measurements with SA water in order to correct the

observed IN concentrations. Second, we chose to use SA water as standard for quantifying instrument uncertainties because we accumulated a considerable number of SA water experiments performed by different users over a longer time period and therefore, we have the best statistics for this sample. In Sect. 3.3 we use the SA water experiments to establish instrumental uncertainties stemming from well-to-well temperature variations.

**Section 4.2: There is no mention on how these suspensions/dilutions are made, how are the particles suspended? This could be particularly important given the results for NX-illite. I recommend adding this information in this section or the methodology.**

Thank you for pointing this out. We have now added that: *An initial stock suspension of 0.1 wt % NX-illite was prepared and then diluted to produce additional mass concentrations of NX-illite of 0.05 and 0.01 wt%. The suspensions were manually shaken for 30 s, poured into a dispensing tray and then immediately pipetted into the well plate. Triplicates of each suspension concentration were investigated with DRINCZ..."* to lines 364-367 in Section 4.2.

**Figure 9: it looks like the temperature intervals where no freezing events were observed are displayed in this figure, i.e. when binning the data, temperature intervals where 0 events were observed are still shown in the cumulative plots. As there are triplicates in this figure it makes it hard to discern which data points are real freezing events and which are artefacts of the binning process. This makes it difficult to interpret the data and the extent of the discrepancy between dilutions. If this is the case, I suggest removing the data points from the cumulative plots where there were no freezing events within a temperature interval for clarity.**

Indeed, the plot shows triplicates for each weight percent but the values are not binned but just plotted at the observed temperature. We agree that the plot is a bit hard to interpret so we have remade the figure to help differentiate between the values reported in literature and shaded the triplicates to more clearly see the run-to-run variability at the different NX-illite concentrations. We have decided to keep all the observed $n_{sBET}$ values since cumulative active site densities indeed remain at a constant value when there is no freezing event in a given time interval. Moreover, constant values indicate that there is a poor data basis relying on rare freezing events.

**Line 355-357: I would not definitively say that the ns is extended as expected. All three triplicates for the 0.01wt% dilutions are giving warming freezing temperatures than the higher weight percent suspensions (at the same value of ns). See major comment.**

The purpose of the sentence is to state that by diluting the solution, we can observe freezing at lower temperatures and measure $n_s$ at higher values. This is true even if the 0.01 wt% is slightly higher in $n_s$ than the 0.05 and 0.1 wt% (within the instrumental uncertainty). Since we agree that we do not show a textbook case for the extension of $n_s$ range by dilution, we have removed "as expected" from the sentence (376-377).

**Line 357-358: In relation to the major comments, you state a few data points from the 0.01wt% suspension appear as outliers (and only at warmer temperatures), whereas all three runs for this dilution are shifted to warmer temperatures for the same value of ns. I suggest restructuring this paragraph to better represent the data and discuss the inconsistencies.**

We have removed "a few data points" from the sentence to more accurately represent the higher $n_{sBET}$ of the lowest weight percent solution at warm temperatures. As described in the response to the major comment, this discrepancy may be due to the presence of a few random active sites which lead to an increase in $n_{sBET}$ that extends to lower temperatures or

issues arising from diluting from a stock suspension. We have reworded lines 377-382 to:
*"Similar to the observations of Harrison et al. (2018), the data points from the 0.01 wt. % solution appear as outliers at the warmest temperatures. However, it is not possible to determine if these outliers are due to random freezing events that occur at high temperatures and therefore produce elevated cumulative $n_{sBET}$ values at lower temperatures or if they are due to an uneven distribution of the active sites in each aliquot that may result from diluting a single stock suspension rather than producing individual weight percent suspensions (Harrison et al., 2018)."* to offer an explanation for the observed divergence in the $n_{sBET}$ between the wt% suspensions.

**Line 358-360: In relation to the above, you reference that Harrison et al. (2018) used individually weighed suspensions rather than a single stock suspension to minimise the effect of uneven particle distributions. Why was this not done here if you believe this is the issue? This seems important as you are validating the instrument yet have inconsistent results on dilution.**

As the discrepancies in $n_{sBET}$ fall within the instrumental uncertainty of DRINCZ, looking for the true cause for the observed differences in the $n_{sBET}$ values after dilution might be an over-evaluation of the data. Moreover, in addition to instrument uncertainties, there is also random variability in *FF* curves obtained from drop freezing assays. Indeed, when examining the *FF* curves shown in Fig. A4, there is considerable variability between triplicates performed with the same suspension concentration. Additionally, there are very few freezing events that occur at the highest temperatures in the 0.01 wt% suspension. Therefore, these high temperature freezing events that are responsible for the high cumulative $n_{sBET}$ values of the 0.01 wt% suspension shown in Fig.9 can be random.

**Line 362-363: At temperatures colder than -15 ˚C this doesn't seem to be the case (especially if you look at the 0.1-0.05wt% suspensions). There is just as good agreement with BINARY at colder temperatures (Hiranuma et al., 2015) but no comparison is made to this instrument.**

We have now changed the sentence to state that the data falls between BINARY Leeds-NIPI and IR-NIPI as follows: *"Furthermore, considering the ± 0.9 ˚C uncertainty, depicted by the horizontal error bars, the differences between concentrations are not significant. They fall within the same range as the measurements of Beall et al, (2017) and between BINARY and Leeds-NIPI and IR-NIPI at colder temperatures (Fig. 9)."* (lines 384-385)

**Line 374-375: Were the samples analysed at this field location or in the lab where the background freezing has been characterised? Were blank (pure SA water) experiments run at the time of these experiments to check the background signal had not changed?**

Thank you for pointing this out. The samples were actually measured in the laboratory in Zurich and we have now clarified this by adding this information to lines (403-404): *"The samples were shipped and stored frozen until processed with DRINCZ at the Atmospheric Physics Laboratory at ETH Zurich to avoid any bacterial growth or changes due to liquid storage (Stopelli et al., 2014)."* Furthermore, we always ran an SA water blank before running DRINCZ on a measurement day to ensure that the background is the same and the system is working properly.

**Technical comments**

**Line 41-43: This sentence needs restructuring/ re-wording as it is a bit clunky. E.g. an ice nucleating particle (singular) cannot get immersed in multiple cloud droplets.**

Thank you, we have now made the sentence singular

**Line 54: Should the word 'or' be in this sentence?**

We have removed "or"

**Line 57-59: No available technique can detect the lowest INP concentrations that are actually present in the atmosphere. I would suggest putting in a range of the INP concentrations detected with these techniques and rewording to say "to detect lower atmospheric INP concentrations".**

We have now reworded the sentence to state: *"lower atmospheric INP concentrations."* (line 60). We have decided not to include a range as the measurable INP concentrations depend on the sampling method (e.g. time of sampling, impinger, filter etc.) as well as the measurement technique.

**Line 102: What material is the 96-well plate made from? Polypropylene? I suggest adding here.**

Thank you, we have now added polypropylene to the text.

**Line 170-174: Suggest removing the terms 'potential' and 'possible' as adding 0 ˚C ethanol to ethanol at -30 ˚C will create a gradient, even if only small. Cooling the ethanol to 0 ˚C simply minimises this gradient.**

Done

**Line 201-201: Consider rephrasing this sentence.**

We have now added a reference to Fig. 3 to clarify that the spread is referring to the temperature calibration.

**Line 208-209: Consider rephrasing for ease of understanding.**

We have now clarified that the observed bias is referring to the freezing temperature bias across the well plate.

**Line 235-236: Harrison et al. 2018 is not a suitable reference in this instance. As I understand, they make individual temperature measurements for each well and as such, they take into account the horizontal gradient in temperature across the plate without the need for such a correction.**

The Harrison et al. (2018) citation here is just meant as an example that such gradients do exist in block-based systems, which are observed in the IR-NIPI. Therefore we have changed the sentence (lines 251-252) to reflect this by adding *"… have been observed or modelled."*

**Figure 2c: Perhaps label the peak which signifies initial nucleation**

We have now clarified this in the figure caption by adding the sentence: *"The most intense peak corresponds to the ice nucleation temperature and the second most intense peak is due to the slow freezing of the solution after nucleation."*

**Line 295: device not devices**

Done

**Line 307-309: This representation of the background you present is for DRINCZ in this particular lab environment. The baseline may change in field studies. I suggest rephrasing this section.**

We have now added the preceding sentence (lines 323-324): *"Furthermore, an SA water sample is run as a standard at the beginning of each measurement day to ensure the system is operating correctly."* as to further motivate the use of SA water as a background and to ensure that the instrument background is reproducible in other settings.

**Line 356: Suggest changing to "samples overlap to an extent"**

Done

**Line 445: missing bracket**

Thank you.

**Figure 9: the triangular symbols are hard to distinguish from one another. Suggest using different symbol shapes.**

Thank you for pointing this out. We have now updated the symbols for clarity.

**References:**

Beall, C. M., Stokes, M. D., Hill, T. C., DeMott, P. J., DeWald, J. T. and Prather, K. A.: Automation and heat transfer characterization of immersion mode spectroscopy for analysis of ice nucleating particles, Atmos Meas Tech, 10(7), 2613–2626, doi:10.5194/amt-10-2613-2017, 2017.

Harrison, A. D., Whale, T. F., Rutledge, R., Lamb, S., Tarn, M. D., Porter, G. C. E., Adams, M. P., McQuaid, J. B., Morris, G. J. and Murray, B. J.: An instrument for quantifying heterogeneous ice nucleation in multiwell plates using infrared emissions to detect freezing, Atmospheric Meas. Tech., 11(10), 5629–5641, doi:https://doi.org/10.5194/amt-11-5629-2018, 2018.

Stopelli, E., Conen, F., Zimmermann, L., Alewell, C. and Morris, C. E.: Freezing nucleation apparatus puts new slant on study of biological ice nucleators in precipitation, Atmospheric Meas. Tech., 7(1), 129–134, doi:10.5194/amt-7-129-2014, 2014.